

# The Berkeley High Resolution Tropospheric NO₂ Product

Joshua L. Laughner[1], Qindan Zhu[2], and Ronald C. Cohen[1,2]

[1]Department of Chemistry, University of California, Berkeley, Berkeley, CA 94720
[2]Department of Earth and Planetary Sciences, University of California, Berkeley, Berkeley, CA 94720

**Correspondence:** Ronald C. Cohen (rccohen@berkeley.edu)

**Abstract.** We describe upgrades to the Berkeley High Resolution (BEHR) $NO_2$ satellite retrieval product. BEHR v3.0 builds on the NASA version 3 standard Ozone Monitoring Instrument (OMI) tropospheric $NO_2$ product to provide a high spatial resolution product that is consistent with daily variations in the 12 km a priori $NO_2$ profiles. Other improvements to the BEHR v3.0 product include surface reflectance and elevation, and factors affecting the $NO_2$ a priori profiles such as lightning and anthropogenic emissions.

We describe the retrieval algorithm in detail and evaluate the impact of changes to the algorithm between v2.1C and v3.0B has on the retrieved $NO_2$ vertical column densities (VCDs). Not surprisingly, we find that, on average, the changes to the a priori $NO_2$ profiles and the update to the new NASA slant column densities have the greatest impact on the retrieved VCDs. More significantly, we find that using daily a priori profiles results in greater average VCDs than using monthly profiles in regions and times with significant lightning activity.

The BEHR product is available as four subproducts on the University of California DASH repository: using monthly a priori profiles at native OMI pixel resolution (https://doi.org/10.6078/D1N086) and regridded to $0.05° \times 0.05°$ (https://doi.org/10.6078/D1RQ3G) and using daily a priori profiles at native OMI (https://doi.org/10.6078/D1WH41) and regridded (https://doi.org/10.6078/D12D5X) resolutions.

## 1 Introduction

Nitrogen oxides ($NO + NO_2 \equiv NO_x$) are trace gases in the atmosphere and are key species controlling air quality and affecting radiative balance. $NO_x$ regulates the chemical production of tropospheric ozone (Jacob et al., 1993), which affects the radiative balance in the upper troposphere (Myhre et al., 2013) and is harmful to plants (Haagen-Smit et al., 1952; Heath, 1975), animals, and humans (Menzel, 1984) at the surface. $NO_x$ directly affects the radiative balance of the atmosphere (e.g. Kiehl and Solomon, 1986). It also plays a role in the formation of aerosol particles (Izumi and Fukuyama, 1990; Pandis et al., 1992; Carlton et al., 2009; Rollins et al., 2012), which also affect the radiative balance of the atmosphere (Boucher et al., 2013). Exposure to fine particles is also a strong factor controlling life expectancy (Pope et al., 2009). Additionally, $NO_x$ itself is harmful (Kagawa, 1985; Chauhan et al., 1998; Wegmann et al., 2005; Kampa and Castanas, 2008).

$NO_x$ is emitted from a variety of sources, both anthopogenic and natural. Anthropogenic sources typically involve combustion, including motor vehicles and fossil fuel electrical generation. Natural sources include biomass burning, lightning, and soil



bacteria. Understanding all of these sources is crucial to understanding the reactive nitrogen budget and predicting how future changes in emissions will affect air quality and climate change.

Satellite observations provide uniquely comprehensive spatial maps of $NO_2$, allowing inference of $NO_x$ emissions. Early instruments (i.e. the Global Ozone Monitoring Experiment, GOME and GOME-2, and the SCanning Imaging Absorption

SpectroMeter for Atmospheric CHartographY, SCIAMACHY) allowed inferences at the scale of entire continents or entire metropolitan regions, including cities and their surroundings. More recent instruments (the Ozone Monitoring Instrument, OMI, and Tropospheric Monitoring Instrument, TROPOMI) have much higher resolution, allowing inferences about individual point sources and urban cores. Ground based measurements sample emissions at specific points in great detail; however, extrapolating such measurements to an entire region requires assumptions that are difficult to test, such as fleet composition

and operating mode (e.g. Fujita et al., 2012; Anderson et al., 2014), that can bias estimates of the total vehicle emissions from a region. Satellite observations cannot currently provide the same level of detail as a roadside measurement, but by observing the entire city, provide a top-down constraint on its total $NO_x$ emissions that include observations on every point in the domain. Satellite observations have been used in a wide variety of applications in this vein, including direct observation of emissions and trends (e.g. Russell et al., 2012), plume analysis to derive emissions and chemical lifetime (e.g. Beirle et al., 2011; Valin

et al., 2013; Lu et al., 2015; Liu et al., 2016, 2017), model constraint (e.g. Travis et al., 2016), and data assimilation (e.g. Miyazaki et al., 2012, 2017).

Satellite measurements have also been used to constraint natural $NO_x$ sources as well, predominantly biomass burning (e.g. Mebust et al., 2011; Huijnen et al., 2012; Mebust and Cohen, 2013, 2014; Bousserez, 2014; Schreier et al., 2014; Castellanos et al., 2015; van Marle et al., 2017), lightning (e.g. Beirle et al., 2004; Martin et al., 2007; Beirle et al., 2010; Bucsela et al.,

2010; Miyazaki et al., 2014; Pickering et al., 2016; Nault et al., 2017), and soil $NO_x$ (e.g. van der A et al., 2008; Hudman et al., 2010, 2012; Zörner et al., 2016). The episodic and geographically disparate nature of these sources (especially lightning and biomass burning) make satellite observations an ideal method to constrain their emissions, given satellites' continuous data record and broad geographic coverage.

The absorption of NO is in the UV, making it too difficult to observe. In contrast, the visible absorbance of $NO_2$ is strong

and inferences about total $NO_x$ are made from $NO_2$ measurements. For a measurement of tropospheric $NO_2$, several steps are required. First, a UV-visible spectrometer records geolocated solar reflectances from the Earth's surface and a reference spectrum of the sun. Then, absorbances in backscattered sunlight are fit using differential optical absorption spectroscopy (DOAS) or a similar technique to yield a total slant column density (SCD). This quantity represents the amount of $NO_2$ per unit area, integrated along all light paths that reach the detector (Boersma et al., 2001; Richter and Wagner, 2011). Next, the

tropospheric and stratospheric $NO_2$ columns are separated. There are several approaches; some examples include using a data assimilation system to constrain modeled stratospheric columns (Boersma et al., 2011) and an iterative process assuming that areas known a priori to have little tropospheric $NO_2$ are all stratospheric $NO_2$ and interpolating to fill in polluted areas (Bucsela et al., 2013). Finally, the tropospheric SCD is converted into a vertical column density (VCD) in order to account for pixel-to-pixel differences in path length and sensitivity to $NO_2$. The conversion of factor from the SCD to the more geophysically



relevant and easily understood VCD is the air mass factor (AMF, Palmer et al., 2001; Burrows et al., 1999; Slusser et al., 1996; McKenzie et al., 1991).

An AMF is computed by simulating an SCD and VCD for each retrieved pixel. Typically, an a priori $NO_2$ profile is simulated with a chemical transport model (CTM) such as GEOS-Chem, WRF-Chem, the GMI-CTM, or TM4. The modeled VCD can

be calculated by integrating this profile over the troposphere. The modeled SCD requires a radiative transfer model, such as TOMRAD, SCIATRAN, VLIDORT, etc., in combination with the a priori $NO_2$ profile in order to compute the light absorbed by $NO_2$ and thus the SCD that yields that absorbance.

The radiative transfer calculations also require a priori inputs: the sun-satellite geometry, surface reflectance, and surface elevation are all necessary. Knowledge of the cloud and aerosol properties in the pixel is also necessary to account for their

effects on light scattering in the radiative transfer calculations. Aerosol effects are often assumed to be implicitly accounted for in cloud properties (e.g. Boersma et al., 2011) but have been treated explicitly by some retrievals (e.g. Lin et al., 2015).

The accuracy of these input data has a significant impact on the accuracy of the retrieved columns. Lorente et al. (2017) compared seven retrievals and found that input assumptions were responsible for a 42% structural uncertainty in AMFs over polluted areas. A key concern is the resolution of the input data. CTMs are computationally expensive, requiring a trade-off

between spatial and temporal resolution and domain size. For global retrievals, model resolutions of $3° \times 2°$ (Boersma et al., 2011) to $1° \times 1°$ (Krotkov et al., 2017) are typical. Russell et al. (2011) found that increasing the resolution of the $NO_2$ profiles to 4 km altered the retrieved VCDs by up to 75%, primarily by capturing the urban-rural gradient in surface $NO_2$ concentrations. McLinden et al. (2014) found that increasing the a priori profiles' resolution to 15 km resulted in a factor of 2 increase in $NO_2$ column over the Canadian oil sands. Laughner et al. (2016) examined the effect of the profiles' temporal

resolution, and identified up to 40% changes in individual VCDs using day-to-day $NO_2$ profiles compared to monthly averaged profiles. The current trade off to obtain such high resolution profiles is that the retrieval is restricted to a region of the world.

The Berkeley High Resolution (BEHR) Ozone Monitoring Instrument (OMI) $NO_2$ retrieval is one such regional retrieval that provides tropospheric $NO_2$ VCDs over the continental United States using high resolution a priori inputs. Here we describe the updates from v2.1C to v3.0B. There are eight primary changes:

1. Updated to use the v3.0 NASA tropospheric SCDs

2. Surface reflectance updated from version 5 MODIS black sky albedo to version 6 MODIS BRDF product

3. A more physically intuitive visible-only AMF calculation was implemented (the standard total tropospheric AMF calculation is unchanged)

4. New a priori $NO_2$ profiles, with specific changes:

(a) Lightning $NO_2$ included

(b) Monthly profiles use 2012 emissions

(c) Daily profiles used for as many years as possible



5. Temperature profiles taken from WRF-Chem instead of the previous coarse climatology

6. A new gridding method was implemented

7. A variable tropopause height derived from WRF simulations replaced the previous fixed 200 hPa tropopause in the AMF calculations.

8. Surface pressure calculation was changed to follow Zhou et al. (2009) using GLOBE terrain elevation and WRF surface pressure

In this paper, we describe each change in detail and examine the effect of each individual change on the retrieved VCDs. Changes implemented in v3.0A are described first, followed by those implemented in v3.0B. A separate paper validating the new product is in preparation.

## 2 Methods: BEHR

### 2.1 NO$_2$ VCD calculation

The BEHR product calculates tropospheric vertical column densities (VCDs) starting from the tropospheric slant column densities (SCDs) from the NASA Standard Product, v3.0 (Krotkov et al., 2017; Krotkov and Veefkind, 2016), by:

$$V_{\mathrm{BEHR}} = \frac{S_{\mathrm{NASA}}}{A_{\mathrm{BEHR}}} \tag{1}$$

where $V_{\mathrm{BEHR}}$ and $S_{\mathrm{NASA}}$ are the BEHR VCD and NASA SCD, respectively, and $A_{\mathrm{BEHR}}$ is a custom tropospheric air mass factor (AMF), computed with

$$A_{\mathrm{BEHR}} = \frac{(1-f) \int_{p_{\mathrm{surf}}}^{p_{\mathrm{trop}}} w_{\mathrm{clear}}(p)g(p)\,dp + f \int_{p_{\mathrm{cloud}}}^{p_{\mathrm{trop}}} w_{\mathrm{cloudy}}(p)g(p)\,dp}{\int_{p_{\mathrm{surf}}}^{p_{\mathrm{trop}}} g(p)\,dp} \tag{2}$$

where $f$ is the cloud radiance fraction, and $w_{\mathrm{clear}}$ and $w_{\mathrm{cloudy}}$ are the scattering weights for clear and cloudy subscenes (i.e. parts of the pixel), respectively, $p_{\mathrm{trop}}$ is the tropopause pressure, $p_{\mathrm{surf}}$ is the ground surface pressure, $p_{\mathrm{cloud}}$ is the cloud optical centroid pressure, and $g(p)$ is the NO$_2$ a priori profile (Sect. 2.6).

This method produces VCDs that include an estimated below cloud component, and thus can be considered a total tropospheric column. This is desirable for applications focusing on near-surface NO$_2$. Other applications (e.g. cloud slicing) benefit from having a "visible-only" tropospheric AMF that only retrieves NO$_2$ above the cloud in a cloudy subscene. For these "visible-only" AMFs, Eq. (2) is replaced with:

$$A_{\mathrm{BEHR,vis}} = \frac{(1-f) \int_{p_{\mathrm{surf}}}^{p_{\mathrm{trop}}} w_{\mathrm{clear}}(p)g(p)\,dp + f \int_{p_{\mathrm{cloud}}}^{p_{\mathrm{trop}}} w_{\mathrm{cloudy}}(p)g(p)\,dp}{(1-f_g) \int_{p_{\mathrm{surf}}}^{p_{\mathrm{trop}}} g(p)\,dp + f_g \int_{p_{\mathrm{cloud}}}^{p_{\mathrm{trop}}} g(p)\,dp} \tag{3}$$



where $f_g$ is the geometric cloud fraction. The numerator is the same as in Eq. (2); in both cases representing a modeled slant column density. The denominator Eq. (2) is the total modeled tropospheric column, while in Eq. (3) it is only the visible modeled column. Replacing $A_{\mathrm{BEHR}}$ in Eq. (1) with $A_{\mathrm{BEHR,vis}}$ yields a visible-only $NO_2$ column as the output.

The scattering weights ($w_{\mathrm{clear}}$ and $w_{\mathrm{cloudy}}$) are computed from the same look-up table (LUT) as the NASA SP v2.1 and

v3.0 (Bucsela et al., 2013; Krotkov et al., 2017). The scattering weights depend on the solar zenith angle (SZA, $\theta_S$), viewing zenith angle (VZA, $\theta_V$), relative azimuth angle (RAA, $\phi_R$), surface reflectance (Sect. 2.2), and surface pressure (Sect. 2.3). A vector of scattering weights is looked up using 5D multilinear interpolation to obtain the scattering weights for the above input parameters. Note that the RAA is calculated as

$$\phi_{R,\mathrm{tmp}} = |180 + \phi_S - \phi_V| \tag{4}$$

$$\phi_R = \begin{cases} \phi_{R,\mathrm{tmp}} & \text{if} \quad \phi_{R,\mathrm{tmp}} \in [0,180] \\ 360 - \phi_{R,\mathrm{tmp}} & \text{if} \quad \phi_{R,\mathrm{tmp}} > 180 \end{cases} \tag{5}$$

where $\phi_S$ and $\phi_V$ are the solar and viewing azimuth angles, respectively, defined in degrees, and $\phi_{R,\mathrm{tmp}}$ is a temporary variable. The extra factor of 180 in Eq. (4) accounts for the RAA definition used in the scattering weight look up table (where $\phi_R = 0$ indicates that the satellite is opposite the sun, i.e. in the forward scattering position), while Eq. (5) ensures that $\phi_R$ is between 0°and 180°.

A temperature correction, $\alpha(p)$ is applied to the scattering weights interpolated from the look-up table, such that $w(p)$ in Eqs. (2) and (3) is equal to $\alpha(p)w_0(p)$, where $w_0(p)$ is the pressure-dependent scattering weights from the look-up table and $\alpha(p)$ is

$$\alpha(p) = 1 - 0.003 \cdot (T(p) - 220) \tag{6}$$
$$\alpha(p) \in [0.1, 10] \tag{7}$$

where Eq. (7) indicates that $\alpha(p)$ is constrained to the range 0.1 to 10. $T(p)$ is a temperature profile taken from the same WRF-Chem simulation as the $NO_2$ a priori profiles (Sect. 2.6).

## 2.2 Surface reflectivity

### 2.2.1 Over land

BEHR v3.0B uses a bidirectional reflectance factor (BRF) to represent surface reflectivity over land. The BRF is given by

(Stahler et al., 1999)

$$R(\theta_S, \theta_V, \phi_R, \Lambda) = f_{\mathrm{iso}}(\Lambda) + f_{\mathrm{vol}}(\Lambda)K_{\mathrm{vol}}(\theta_S, \theta_V, \phi_R) + f_{\mathrm{geo}}(\Lambda)K_{\mathrm{geo}}(\theta_S, \theta_V, \phi_R) \tag{8}$$



where $R$ is the surface reflectivity, $f_{\mathrm{iso}}$, $f_{\mathrm{vol}}$, and $f_{\mathrm{geo}}$ are coefficients representing the relative contributions of different types of scattering, and $K_{\mathrm{vol}}$ and $K_{\mathrm{geo}}$ are kernels representing the directional dependence of the reflectivity. $\Lambda$ represents a wavelength band, which here is band 3 of the MODIS instrument (459–479 nm).

$K_{\mathrm{vol}}$ is the RossThick kernel (Roujean et al., 1992) and $K_{\mathrm{geo}}$ is the LiSparse kernel (Wanner et al., 1995), corrected to be

reciprocal in $\theta_S$ and $\theta_V$. BEHR calculates both kernels using the formulations given in Stahler et al. (1999). The coefficients, $f_{\mathrm{iso}}$, $f_{\mathrm{vol}}$, and $f_{\mathrm{geo}}$, are taken from the MODIS MCD43D07 (Schaaf, 2015a), MCD43D08 (Schaaf, 2015b), and MCD43D09 (Schaaf, 2015c) BRDF products, respectively. Quality information for these coefficients is obtained from the MCD43D31 product (Schaaf, 2015d). (The combination of these four products will henceforth be referred to as MCD43Dxx.) These products represent a 16-day average; in version 006 (used here), the file date is in the middle of that 16-day averaging window. BEHR

uses the file dated for the day being retrieved for the BRF coefficients.

An average surface reflectance for a given OMI pixel is calculated by computing $R$ for each set of MCD43Dxx coefficients within the bounds of the pixel given by the FoV75 corners from the OMPIXCOR product (Kurosu and Celarier, 2010) and using the SZA, VZA, and RAA of the pixel as inputs to the kernels. All values of $R$ from MCD43Dxx coefficients with non-fill quality flags are averaged to produce the overall surface reflectance for the pixel; however, since coefficients with quality 3 are

significantly lower quality than quality 0 to 2, if the average quality of all MCD43Dxx coefficients within the OMI pixel is $\geq 2.5$, the pixel is flagged as low quality. The pixel is also flagged if $\geq 50\%$ of the MCD43Dxx coefficients have a fill value for the quality (see Sect. A2).

### 2.2.2 Over water

The MCD43Dxx products do not contain coefficients over deep water; therefore, an alternate measure of surface reflectance is

needed. We use the University of Maryland land map (ftp://rsftp.eeos.umb.edu/data02/Gapfilled/Land_Water_Mask_7Classes_ UMD.hdf, accessed 28 Nov 2017) to classify OMI pixels as land or water. Land classes 0 (shallow ocean), 6 (moderate or continental ocean), and 7 (deep ocean) are considered ocean; all others are considered land. The mask is given at 30 arc second resolution; if $> 50\%$ of the mask data points within the FoV75 bounds of the OMI pixel are ocean, the pixel is treated with an ocean surface reflectance.

Ocean surface reflectance is parameterized by SZA using output from the Coupled Ocean Atmosphere Radiative Transfer (COART) model (Jin et al., 2006, hosted at https://satcorps.larc.nasa.gov/jin/coart.html, accessed 2 Mar 2018). The ratio of upwelling to downwelling radiation was simulated for solar zenith angles between $0°$ and $85°$ at $5°$ increments. Additional settings are given in Table 1. The ratio of upwelling to downwelling radiation is linearly interpolated to the SZA of the OMI pixel, and that interpolated ratio is taken as the surface reflectance of the ocean pixel.



| | |
|---|---|
| Wavelength | 430 nm (v3.0A), 460 nm (v3.0B) |
| Atmospheric profile | MLS |
| Boundary layer aerosol model | MODTRAN Maritime |
| Stratospheric aerosol model | Background stratosphere |
| Total aerosol loading | AOD at 500 nm = 1 |
| Wind speed | 5 m s$^{-1}$ |
| Ocean depth | 100 m |
| Chlorophyll | 0.2 mg m$^{-3}$ |
| Ocean particle scattering | Petzold Average, $bb/b = 0.0183$ |
| Bottom surface albedo | 0.1 |

**Table 1.** Additional settings for the COART model used to simulate ocean reflectivity.

## 2.3 Surface pressure

The surface elevation of each OMI pixel is computed by averaging all surface elevation values from the Global Land One-kilometer Base Elevation (GLOBE) database (Hastings and Dunbar, 1999) within in FoV75 bounds of the pixel. In BEHR versions prior to v3.0B, this is converted to a surface pressure with

$$p = (1013.25\ \text{hPa})e^{-z/7400\ \text{m}} \tag{9}$$

where $z$ is the average surface elevation in meters. From v3.0B on, pixel surface pressure is calculated using the method recommended by Zhou et al. (2009):

$$p = p_{\text{WRF}} \left( \frac{T_{\text{WRF}}}{T_{\text{WRF}} + \Gamma \cdot (h_{\text{WRF}} - h_{\text{GLOBE}})} \right)^{-g/R\Gamma} \tag{10}$$

where $p$ is the pixel surface pressure, $p_{\text{WRF}}$, $T_{\text{WRF}}$, and $h_{\text{WRF}}$ are the surface pressure, temperature, and elevation from the WRF model, $h_{\text{GLOBE}}$ is the averaged GLOBE surface elevation, $g$ is gravitational acceleration (9.8 m s$^{-2}$), $R$ is the gas constant for dry air (287 J kg$^{-1}$ K$^{-1}$) and $\Gamma$ the lapse rate (0.0065 K m$^{-1}$).

## 2.4 Tropopause Pressure

BEHR v3.0A and prior versions used a fixed tropopause pressure (200 hPa), BEHR v3.0B utilizes thermal tropopause pressure derived from the temperature profile from the same WRF-Chem simulation as the NO$_2$ a priori profiles. The thermal tropopause is defined as the lowest level at which the average lapse rate between this level and all higher levels within 2 km does not exceed 2 K km$^{-1}$ by World Meteorological Organization (1957). The calculation operationally works in most regions, however, occasionally a discontinuity occurs between adjacent pixels where both pixels approach the 2 K km$^{-1}$ threshold at the same



model level but only one exceeds the threshold at that level. As this discontinuity is only due to the choice of the standard threshold for lapse rate in the criteria, an additional filtering is implemented to identify pixels with abrupt transition in calculated tropopause pressure. New tropopause pressures for these pixels are derived by linear interpolation of tropopause pressures from the nearest valid pixels after filtering.

## 2.5 Cloud products

BEHR v3.0B contains several cloud fraction products: an OMI-derived geometric and radiance cloud fraction, a geometric cloud fraction derived from the Aqua MODIS instrument, and cloud pressure from the OMI $O_2$-$O_2$ algorithm (Acarreta et al., 2004). The OMI-derived quantities are the same as those in the NASA SP v3.0. The MODIS cloud product used is MYD06_L2 (Platnick et al., 2015). As with the MODIS BRDF product, all values of cloud fraction given in MYD06_L2 within each OMI pixels bounds defined by the FoV75 pixel corners are averaged to yield the MODIS-derived cloud fraction for that OMI pixel. Unlike the BRDF product, only Level 2 MODIS granules with times between the start and end time of the current OMI orbit are used.

## 2.6 A priori profiles

### 2.6.1 WRF-Chem configuration

$NO_2$ and temperature a priori profiles are generated using version 3.5.1 of WRF-Chem (Grell et al., 2005) run at 12 km resolution across the continental United States (Fig. S1). The North American Regional Reanalysis (NARR) dataset is used to drive the meteorological initial and boundary conditions, as well as four-dimensional data analysis (FDDA) nudging (Liu et al., 2006). U and V winds, as well as temperature and water vapor are nudged at all levels with nudging coefficients of 0.0003 $s^{-1}$.

Anthropogenic emissions are driven by the National Emissions Inventory 2011 (NEI 11) gridded to 12 km resolution. Each years' emissions are scaled by the ratio of that years total annual emissions to 2011 emission. These total emissions are provided by the Environmental Protection Agency (EPA, 2016). Biogenic emissions are driven by the Model of Emissions of Gases and Aerosols from Nature (MEGAN, Guenther et al., 2006). Lightning emissions are driven by the recommended settings in Laughner and Cohen (2017) for a simulation using FDDA nudging.

Chemistry in WRF-Chem is simulated using the RACM2_Berkeley2 mechanism (Zare, in prep), which is based on the Regional Atmospheric Chemistry Mechanism, version 2 (RACM2, Goliff et al., 2013) with updates to alkyl nitrate and nighttime chemistry (Browne et al., 2014; Schwantes et al., 2015) and the inclusion of methylperoxy nitrate (MPN) chemistry (Browne et al., 2011; Nault et al., 2015, 2016).

For model years 2007 and later, output from the Model for Ozone and Related chemical Tracers (MOZART, Emmons et al., 2010) provided by the National Center for Atmospheric Research (NCAR) at https://www.acom.ucar.edu/wrf-chem/mozart. shtml is used, converted to boundary conditions for WRF-Chem using the MOZBC utility. MOZART data is not available for model years 2005–2006; instead, the chemical data is taken from the GEOS-Chem model v9-02, with updates from Nault et al. (2017). These updates are detailed in Sect. S1 of the supplement. GEOS-Chem instantaneous output is sampled every



three hours. This output is transformed into netCDF files for input into the MOZBC utility by use of the gc2moz utility of the AutoWRFChem package (Laughner, 2017).

Each year is simulated with a 1 month spinup at the anthropogenic emissions levels for that year. The year is simulated continuously, without reinitialization. Instantaneous WRF-Chem output is sampled hourly. For 2007, since MOZBC data was

not available for December 2006, boundary conditions for 1 Jan 2007 were repeated for the first 32 days of the simulation (1 Dec 2006 to 1 Jan 2007) to allow the model time to spin up from the initial conditions.

In the BEHR AMF calculation, the profiles are interpolated, with extrapolation, to the same pressures that the scattering weights are defined on. The $NO_2$ mixing ratio profiles are interpolated in log-log space (e.g., $\ln(NO_2)$ given at $\ln(p_{WRF})$ is interpolated to $\ln(p_{std})$, Bucsela et al., 2008). Temperature is interpolated in semilog space ($T$ given at $\ln(p_{WRF})$ is interpolated

to $\ln(p_{std})$), since lapse rates assume a linear relationship between temperature and altitude, and altitude is proportional to $\ln(p)$. Once interpolated, all profiles within the FoV75 bounds of the OMI pixel are averaged to give the profiles used in calculating the AMF.

### 2.6.2   Daily a priori profiles

As recommended in Laughner et al. (2016), we make use of daily profiles where possible. Both $NO_2$ a priori profiles and the

temperature profiles necessary for the scattering weight temperature correction are drawn from the same simulation. WRF-Chem is configured to provide instantaneous output at the top of every hour. In v3.0A, the last WRF-Chem profile before average time of the OMI pixels over the domain is chosen to provide the a priori $NO_2$ and temperature profiles. In v3.0B, the profile closest in time to the average OMI time is used. These profiles are binned to OMI pixels as described in Sect. 2.6.1.

### 2.6.3   Monthly a priori profiles

Given the computational cost in producing daily a priori profiles, we continue to use monthly average profiles as well to cover years for which daily a priori profiles are unavailable. Monthly profiles are generated from 2012 WRF-Chem output. As in Laughner et al. (2016), an average of all available hourly profiles for a given month weighted by weights $w_l$ given by:

$$w_l = 1 - |13.5 - (l/15) - h|$$
$$w_l \in [0,1] \tag{11}$$

where $l$ is the profile longitude and $h$ is the UTC hour of the profile. This formulation gives highest weight to profiles near OMI overpass time (approximately 13:30 local standard time) while smoothly interpolating between adjoining time zones. The appropriate month's profiles are spatially matched to OMI pixels in the same manner as the daily profiles (Sect. 2.6.2).





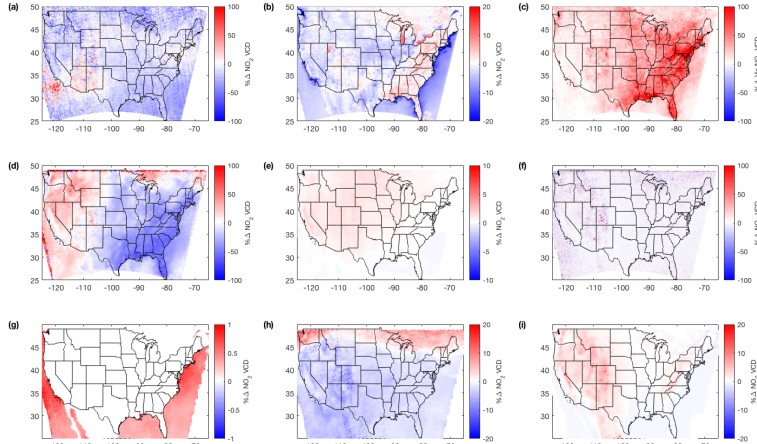

**Figure 1.** Percent change in the tropospheric $NO_2$ column due to each of the algorithm improvements, averaged over Jun–Aug 2012. (c) is for the visible-only column; all others are the total tropospheric column. Changes due to (a) new NASA SCDs, (b) new surface reflectance, (c) new visible AMF calculation, (d) new monthly $NO_2$ profiles, (e) new temperature profiles, (f) new gridding method, (g) change in ocean reflectance LUT from 430 to 460 nm, (h) switch to WRF-derived tropopause pressure, (i) switch to Zhou et al. (2009) surface pressure methodology. Note that the color scale varies among the plots. Averages are for Jun–Aug 2012 and exclude pixels affected by the row anomaly and with cloud fraction $> 0.2$. Monthly average a priori profiles are used for all differences. Wintertime changes and histograms are given in Sect. S4.

## 3 Changes in BEHR v3.0A

### 3.1 NASA v3.0 slant columns

Version 3.0 of the NASA Standard Product introduced a new method of fitting the observed Earthshine radiances to yield total SCDs (Krotkov et al., 2017; Marchenko et al., 2015). This new fitting approach eliminates a positive bias identified
5   by Belmonte Rivas et al. (2014), and reduces the total SCDs retrieved. For much of the globe, this reduction is attributed to the stratospheric SCD, but over the continental US, it is attributed to the tropospheric SCD. Thus, the broad reduction in tropospheric VCDs seen here (Fig. 1a, Tables 2 and 3) due to the new SCD fitting is consistent with Krotkov et al. (2017).

### 3.2 Surface reflectance: BRF

Generally, UV-vis AMFs increase (thus $NO_2$ VCDs decrease) with increasing surface reflectance, due to greater sensitivity to
10   near surface $NO_2$. This pattern is apparent in Fig. 1b and Fig. 2a, as the $NO_2$ VCDs show the expected inverse relationship to the surface reflectance.



| | | JJA | | DJF | |
|---|---|---|---|---|---|
| | | Mean | Median | Mean | Median |
| | SCDs | $-14 \pm 14$ | $-13^{+9}_{-9}$ | $-21 \pm 15$ | $-21^{+9}_{-9}$ |
| | Surf. refl | $-1.5 \pm 2.8$ | $-1.4^{+1.8}_{-1.8}$ | $0.17 \pm 6.87$ | $0.22^{+3.63}_{-4.29}$ |
| | Vis. AMF formulation* | $24 \pm 18$ | $20^{+15}_{-10}$ | $18 \pm 14$ | $15^{+11}_{-8}$ |
| | NO$_2$ profiles | $-9.8 \pm 24.3$ | $-11^{+20}_{-16}$ | $-0.5 \pm 7.7$ | $0.32^{+3.33}_{-4.23}$ |
| Monthly | Temperature profiles | $0.5 \pm 0.4$ | $0.44^{+0.38}_{-0.29}$ | $1.5 \pm 0.9$ | $1.4^{+0.8}_{-0.7}$ |
| | Gridding | $-0.66 \pm 6.65$ | $-0.65^{+4.00}_{-4.05}$ | $-0.58 \pm 10.45$ | $-0.64^{+6.28}_{-6.19}$ |
| | Ocean LUT/profile time** | $0.42 \pm 0.12$ | $0.38^{+0.12}_{-0.05}$ | $0.41 \pm 0.16$ | $0.43^{+0.10}_{-0.11}$ |
| | Variable trop. | $-2.4 \pm 1.5$ | $-2.2^{+0.6}_{-1.0}$ | $1.9 \pm 2.4$ | $2^{+1}_{-1}$ |
| | Hypsometric Surf. Pres | $0.55 \pm 0.87$ | $0.3^{+0.7}_{-0.4}$ | $0.7 \pm 0.9$ | $0.4^{+0.7}_{-0.4}$ |
| | NO$_2$ profiles | $0.86 \pm 20.14$ | $-0.54^{+15.46}_{-12.27}$ | $-1.3 \pm 10.0$ | $-0.033^{+4.693}_{-6.724}$ |
| | Temperature profiles | $0.62 \pm 0.48$ | $0.62^{+0.33}_{-0.38}$ | $1.5 \pm 1.2$ | $1.2^{+1.0}_{-0.6}$ |
| Daily | Gridding | $-0.82 \pm 6.83$ | $-0.84^{+4.15}_{-4.13}$ | $-0.61 \pm 10.58$ | $-0.69^{+6.36}_{-6.25}$ |
| | Ocean LUT/profile time | $0.036 \pm 0.666$ | $0.044^{+0.403}_{-0.406}$ | $-0.084 \pm 0.557$ | $-0.036^{+0.282}_{-0.371}$ |
| | Variable trop. | $-1.9 \pm 2.4$ | $-2.3^{+1.6}_{-1.2}$ | $2.6 \pm 2.6$ | $2.3^{+1.8}_{-1.2}$ |
| | Hypsometric Surf. Pres | $0.61 \pm 1.01$ | $0.36^{+0.86}_{-0.43}$ | $1 \pm 1$ | $0.58^{+1.22}_{-0.60}$ |

**Table 2.** Percent differences in averaged NO$_2$ VCDs for each increment. Means are given with $1\sigma$ uncertainties; medians are given with uncertainties as the distance to the upper and lower quartiles. Outliers were removed before calculating these statistics. *Statistics for visible-only NO$_2$ column. **Statistics only for ocean pixels.

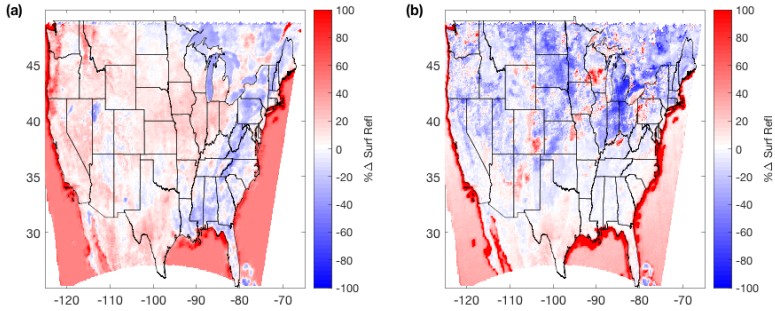

**Figure 2.** Difference in surface reflectance between BEHR v2.1C (MODIS MCD43C3 black-sky albedo, old ocean look up table) and BEHR v3.0B (MODIS MCD43Dxx BRF, new look up table) for (a) summer (JJA) and (b) winter (DJF).



The changes in average surface reflectance are due primarily to the upgrade from version 5 to version 6 of the MODIS product. Figure S4 breaks down the change in surface reflectance due to the upgrade from version 5 to 6 of the MODIS products separately from the change from black-sky to BRF surface reflectance. From Fig. S4a and b, it is evident that the spatial pattern seen in the surface reflectance changes are due primarily to the differences between version 5 and 6. Differences

between version 5 and 6 were listed at https://lpdaac.usgs.gov/dataset_discovery/modis/modis_products_table/mcd43c3_v006 as of 5 Feb 2018. Two improvements of note are:

- Change from a land cover-based backup database to one based on full inversions. Notably, the summertime decreases along the east coast (Fig. 2a, S4a) are somewhat spatially correlated with deciduous broadleaf forest, mixed forest, and woody savanna land cover types that are rare elsewhere in the country (Fig. S5).

- Change from using the majority snow or no-snow status from the 16-day observation window to the current day status. In Fig. S4b, the largest changes are seen sporadically in the northern half of the country, which suggests snow cover is impacting the surface reflectance.

We have not rigorously tested these specific changes as the cause for the spatial pattern of changes in surface reflectance; rather, our point is that the change from version 5 to 6 of the MODIS products is a larger driver of the change in average surface

reflectance than the change from black-sky to BRF. However, Fig. S4e illustrates that the change in individual pixels' surface reflectance due to the switch to a BRF is significant. The switch to a BRF surface reflectance is expected to improve retrieval accuracy of individual pixels and therefore is valuable to users interested in day-to-day variations in $NO_2$ VCDs (Vasilkov et al., 2017).

### 3.3 New visible-only AMF calculation

The formula for the v3.0 visible-only AMF is given in Eq. (3). Conceptually, this is the model SCD divided by the modeled VCD. In v2.1C, an alternate formulation was used:

$$A_{\mathrm{BEHR,vis}} = (1-f)A_{\mathrm{clear,vis}} + fA_{\mathrm{cloudy,vis}} \tag{12}$$

where $f$ is again the cloud radiance fraction and

$$A_{\mathrm{clear,vis}} = \frac{\int_{p_{\mathrm{surf}}}^{p_{\mathrm{trop}}} w_{\mathrm{clear}}(p)g(p)\,dp}{\int_{p_{\mathrm{surf}}}^{p_{\mathrm{trop}}} g(p)\,dp} \tag{13}$$

$$A_{\mathrm{cloudy,vis}} = \frac{\int_{p_{\mathrm{cloud}}}^{p_{\mathrm{trop}}} w_{\mathrm{cloudy}}(p)g(p)\,dp}{\int_{p_{\mathrm{cloud}}}^{p_{\mathrm{trop}}} g(p)\,dp} \tag{14}$$

This earlier method assumes that each pixel can be treated as two totally independent subpixels, one clear and one cloudy. This seems a logical extension of the independent pixel approximation (Cahalan et al., 1994; Marshak et al., 1998), but the physical interpretation is less clear than the new formulation.

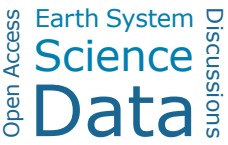

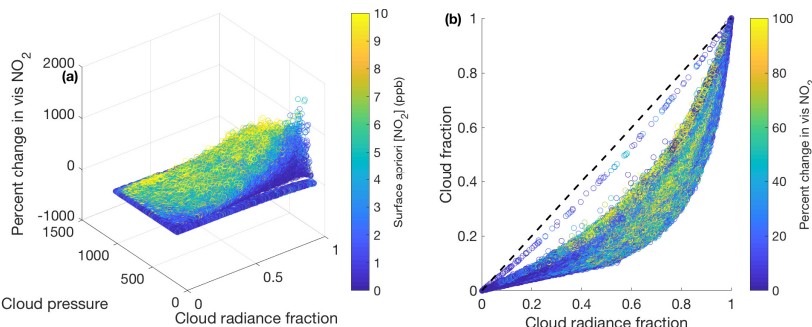

**Figure 3. (a)** The percent change in the visible-only $NO_2$ VCD versus cloud radiance fraction, cloud pressure, and surface $NO_2$ concentration in the a priori profiles. **(b)** The percent change in visible-only $NO_2$ VCD as a function of cloud radiance fraction and geometric cloud fraction. The color scale saturates at 10 ppbv in (a) and 100% in (b) to emphasize the distribution of the percent changes. The black dashed line is the 1:1 line.

Although both approaches are conceptually valid, they are not mathematically identical, and so the retrieved visible tropospheric $NO_2$ column increases between v2.1C and v3.0A (Fig. 1c). The increase approaches 100% over the eastern US, decreasing to 0 towards the west coast.

The reason for this change is how the two approaches weight the clear vs. cloudy contributions to the pixel. In the v2.1C

5   approach (Eqs. 12–14), the visible-only AMF is simply a cloud radiance fraction-weighted average of clear and above-cloud AMFs. Above-cloud AMFs are typically large, as the sensitivity of a remote sensing instrument to an above-cloud column is high both because of the altitude at which the column resides and the highly reflective cloud. As cloud fraction increases, the v2.1C visible AMF is weighted more heavily toward the above-cloud AMF, which leads to smaller retrieved visible $NO_2$ columns.

10   In contrast, the v3.0A visible AMF is the ratio of the modeled SCD to modeled visible VCD. The modeled visible VCD is calculated as the sum of to-ground and above-cloud VCDs weighted by the *geometric* cloud fraction, which tends to be smaller than the radiance cloud fraction. Thus, the v3.0 visible AMF should better account for the fact that the clear part of the pixel scatters less light than the cloudy part of the pixel. This leads to smaller AMFs than the v2.1C formulation, reflecting the fact that fewer photons interact with the clear part of the pixel, especially below the cloud top height.

15   Figure 3 shows the percent difference between the v2.1C and v3.0A visible-only VCDs as a function of several relevant input variables. We note the following patterns:

1. Greater percent difference with lesser cloud pressure

2. Greater percent difference with greater surface $NO_2$ concentration in the a priori profiles

3. Greater percent difference with greater difference between the geometric and radiance cloud fractions



All of these follow the conceptual difference between the two AMF formulations; the v2.1C AMF is retrieving the visible VCDs weighted by the amount of light that interacted with the clear and cloudy parts of the pixel; this will tend to be weighted towards the above-cloud part of the visible VCD. In contrast, the v3.0A AMF is designed to account for the difference between the radiance and geometric cloud fraction. Thus, as the amount of lower troposphere $NO_2$ increases or the gap between the amount of the pixel physically covered by cloud and the fraction of light from the cloud increases, the difference in the visible-only VCDs will be larger. Note in Fig. 3b that as both cloud fractions converge to 0 or 1 the difference in the visible-only VCDs tends towards 0, as expected since both the old and new visible-only AMF formulations reduce to the same expression if $f = f_g = 0$ or $f = f_g = 1$.

### 3.4 New WRF-Chem profiles

#### 3.4.1 Update to new monthly average profiles

There are three significant changes from the old monthly average profiles used in v2.1C and before:

1. Lightning $NO_x$ emissions are included in the profiles; these were not available in WRF-Chem when the previous profiles were simulated.

2. The anthropogenic emissions used now are from the National Emissions Inventory, 2011 (NEI 2011), scaled based on total annual emission to 2012 levels. 2012 boundary conditions and meteorology also used. In v2.1C and earlier, NEI 2005 emissions were used.

3. The chemical mechanism was updated from the Regional Acid Deposition Model, version 2 to the custom mechanism described in Sect. 2.6.1.

The changes in the summer average VCDs due to the profile update is shown in Fig. 1d. The effect of including lightning $NO_x$ emissions is most apparent, causing the $\sim 30\%$ decrease (5th/95th percentiles: 8% and 55%) in VCDs in the SE US (averaged east of 95°and south of 45°). This is due to the increased contribution of UT $NO_2$ to the a priori profiles compared to the v2.1C profiles. As this $NO_2$ is located at higher sensitivity altitudes, the AMF is increased (and the retrieved VCD decreased) to reflect that higher sensitivity.

The increased VCDs along the west coast are caused by changes to the UT $NO_2$ profiles. The UT $NO_2$ over the west decreased compared to the old a priori profiles. This may be due either to the change in chemical mechanism or to a change in the $O_3$ boundary condition, which would affect the simulated UT $NO:NO_2$ ratio.

#### 3.4.2 Daily vs. monthly profiles

Figure 5 shows the difference in v3.0B in the average total tropospheric $NO_2$ columns when using daily $NO_2$ profiles rather than monthly average profiles. Figure 5a is the summer (JJA) average, and shows a significant increase in VCDs along the eastern US, which is not present in the winter (DJF) average (Fig. 5b). The timing and location suggests that this difference



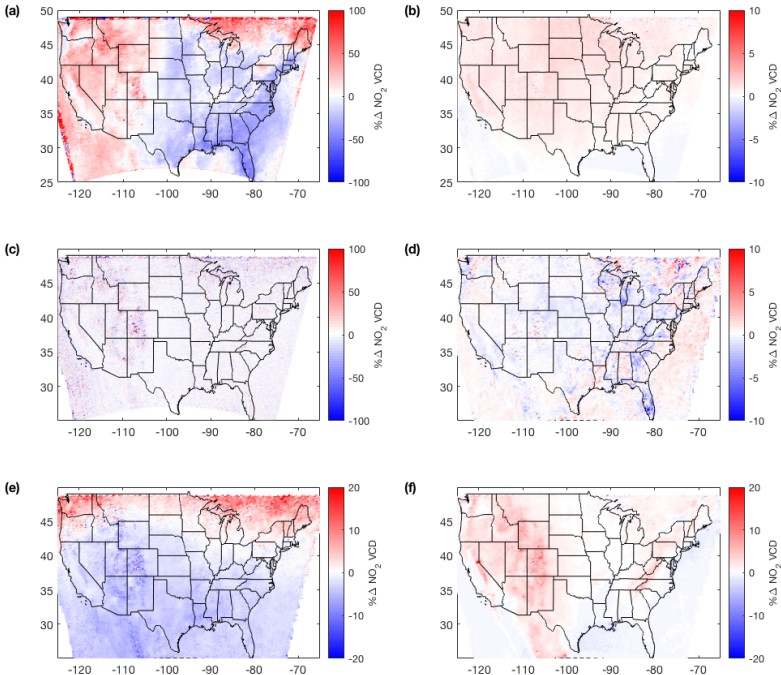

**Figure 4.** Percent change in the total tropospheric $NO_2$ column due to each of the algorithm improvements for the subproduct using daily profiles. Changes due to (a) new $NO_2$ profiles, (b) new temperature profiles, (c) new gridding, (d) change in profile time selection and ocean reflectance LUT from 430 to 460 nm, (e) switch to WRF-derived tropopause pressure, (f) switch to the Zhou et al. (2009) surface pressure methodology. Note that in (a), the difference is against an increment using monthly average profiles; also note that the color scale varies among the plots. Averages are for Jun–Aug 2012 and exclude pixels affected by the row anomaly and with cloud fraction $> 0.2$. Wintertime changes and histograms are given in Sect. S4.

is due to lightning, as the southeast US especially has very active lightning (Laughner and Cohen, 2017; Travis et al., 2016; Hudman et al., 2007).

Since the averages in Fig. 1 and Fig. 5 only use pixels with cloud fraction $\leq 0.2$, a reasonable hypothesis is that the daily profiles for those less-cloudy pixels tend to have less upper tropospheric $NO_2$ than do the monthly average profiles (which include all days regardless of cloudiness). However, this is not the case, as Fig. 6 shows that cloud filtering does not change the distribution of UT $NO_2$ in the a priori profiles. Rather, it is seems to be caused by the order of averaging. The average daily and monthly profiles in the SE US are similar, but the median profiles are quite different (Fig. S3). This is expected, as profiles influence by lightning are less frequent than those not influenced, but the amount of $NO_2$ introduced into the UT by lightning is large, leading to a skewed distribution of UT $NO_2$ concentrations (Fig. 6). When shape factors are considered, it is evident



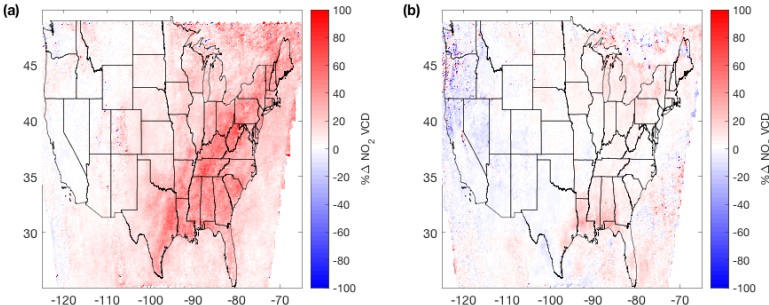

**Figure 5.** (a–b) Percent difference in NO$_2$ VCDs using daily instead of monthly profiles averaged over (a) Jun–Aug and (b) Jan, Feb, Dec 2012. Averages exclude pixels affected by the row anomaly and with cloud fraction $> 0.2$.

| | | JJA | | DJF | |
| | | Mean | Median | Mean | Median |
|---|---|---|---|---|---|
| | SCDs | $-15 \pm 49$ | $-16^{+27}_{-27}$ | $-21 \pm 48$ | $-20^{+24}_{-28}$ |
| | Surf. refl | $-1.6 \pm 4.7$ | $-1.3^{+2.8}_{-3.4}$ | $-0.29 \pm 8.60$ | $0.16^{+5.21}_{-6.07}$ |
| | Vis. AMF formulation* | $6.7 \pm 10.6$ | $0.88^{+10.25}_{-0.88}$ | $-0.052 \pm 0.770$ | $0^{+0}_{-0}$ |
| | NO$_2$ profiles | $-8.3 \pm 25.7$ | $-6.9^{+15.8}_{-19.9}$ | $-2.1 \pm 8.5$ | $-0.95^{+3.59}_{-5.70}$ |
| Monthly | Temperature profiles | $0.49 \pm 0.51$ | $0.44^{+0.40}_{-0.31}$ | $1.2 \pm 1.3$ | $1^{+1}_{-1}$ |
| | Gridding | N/A | N/A | N/A | N/A |
| | Ocean LUT/profile time** | $0.41 \pm 0.17$ | $0.37^{+0.14}_{-0.08}$ | $0.44 \pm 0.29$ | $0.46^{+0.19}_{-0.19}$ |
| | Variable trop. | $-2.2 \pm 1.8$ | $-2.1^{+0.9}_{-1.2}$ | $1.5 \pm 2.8$ | $1.6^{+1.6}_{-1.5}$ |
| | Hypsometric Surf. Pres | $0.49 \pm 0.78$ | $0.26^{+0.70}_{-0.26}$ | $0.69 \pm 0.87$ | $0.32^{+0.90}_{-0.32}$ |
| | NO$_2$ profiles | $1 \pm 25$ | $2.3^{+13.7}_{-15.2}$ | $-2.7 \pm 12.5$ | $-1.4^{+6.2}_{-8.8}$ |
| | Temperature profiles | $0.53 \pm 0.91$ | $0.43^{+0.66}_{-0.53}$ | $1.1 \pm 1.7$ | $0.87^{+1.27}_{-0.98}$ |
| Daily | Gridding | N/A | N/A | N/A | N/A |
| | Ocean LUT/profile time | $0.092 \pm 0.613$ | $0^{+0}_{-0}$ | $0.053 \pm 0.573$ | $0^{+0}_{-0}$ |
| | Variable trop. | $-1.9 \pm 2.9$ | $-1.8^{+1.5}_{-1.8}$ | $2.1 \pm 3.7$ | $1.7^{+2.6}_{-1.6}$ |
| | Hypsometric Surf. Pres | $0.53 \pm 0.91$ | $0.24^{+0.88}_{-0.24}$ | $0.5 \pm 0.9$ | $0.081^{+0.893}_{-0.081}$ |

**Table 3.** Percent differences in individual pixels' NO$_2$ VCDs for each increment. Means are given with $1\sigma$ uncertainties; medians are given with uncertainties as the distance to the upper and lower quartiles. Outliers were removed before calculating these statistics. *Statistics for visible-only NO$_2$ column. **Statistics only for ocean pixels.





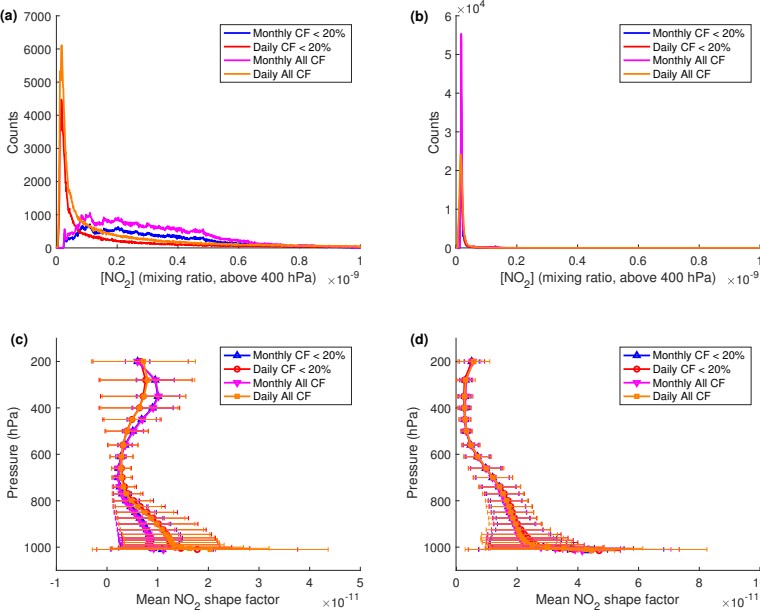

**Figure 6.** (a–b) Frequency distribution of average $NO_2$ above 400 hPa in the a priori profiles for the southeast US (a) and northwest US (b), from Jun–Aug 2012. (c–d) Mean a priori $NO_2$ shape factors over the southeast US (c) and northwest US (d) for Jun–Aug, 2012. The error bars are $\pm 1\sigma$. In all plots, the red and blue lines are only profiles from pixels with cloud fraction $\leq 20\%$, the purple and orange lines use all pixels. The regions (SE and NW US) are shown in Fig. S2.

that this skewed distribution causes monthly shape factors to place more weight on UT $NO_2$ than do the daily shape factors (Fig. 6c).

## 3.5 WRF-Chem temperature profiles

Simulated or recorded temperature profiles are necessary to correct for the temperature dependence of the $NO_2$ cross section
5 (Sect. 2.1 of this paper, also Bucsela et al., 2013). BEHR v2.1C used temperature profiles provided by NASA at $5° \times 2°$
resolution. Recently, an error was identified in the temperature profile lookup used in BEHR v2.1C. Correcting this error
changes the v2.1C VCDs by $-1.7 \pm 3.8\%$ (summer, $-0.9 \pm 11.2\%$ winter, Fig. S6). Therefore the impact was small in both
seasons, but more variable in the winter.

BEHR v3.0A uses temperature profiles from WRF-Chem at 12 km resolution instead. The effect on total tropospheric VCDs
10 is shown in Fig. 1e. It is also small, $0.5 \pm 0.4\%$ on average in summer using monthly average profiles. Using daily temperature
profiles, the change is slightly more variable ($0.6 \pm 0.5\%$). Therefore, high resolution temperature profiles are significantly less
important than $NO_2$ profiles, which is expected, as temperature should not vary as rapidly in space as $NO_2$.



## 3.6  Gridding method

BEHR v2.1C used a constant value method (CVM) gridding algorithm to oversample the native pixel data to a fixed $0.05° \times 0.05°$ grid. A constant value method assigns the VCD of a given pixel to any grid points within the pixel bounds; this works well when the grid resolution is significantly finer than the native pixel resolution. It was found that the existing algorithm was

at times overly conservative, and did not assign values to grid cells near the border of two pixels.

We tested the two gridding algorithms described in Kuhlmann et al. (2014), a different CVM algorithm and the parabolic spline method (PSM), with updates from Schütt (2017). The PSM attempts to recover maxima in $NO_2$ between adjacent pixels by fitting the $NO_2$ VCDs with 2D splines and sampling the grid points along those splines. While this algorithm should be an ideal match with our retrieval (as our high resolution profiles are able to better resolve urban-rural $NO_2$ gradients), two

technical challenges persisted. First, non-physical oscillations in the $NO_2$ VCDs would appear, especially on the edge of the row anomaly. Second, in one test, the PSM algorithm resulting in much greater VCDs than the CVM algorithm over a large area. As this is not the expected behavior, v3.0A uses the new CVM method from Kuhlmann et al. (2014). Both the PSM and CVM algorithms are adapted from those available at https://github.com/gkuhl/omi.

Figures 1f and 4c shows the percent change in the VCDs resulting from the change in gridding method. The average effect is

small and no spatial pattern is evident, as would be expected, although individual effects are quite variable (2). The new CVM algorithm correctly assigns grid cells near the border of two pixels to one or the other. If two pixels overlap, an average of their values weighted by the inverse of their area (FoV75Area from the OMPIXCOR product) is assigned.

## 4  Changes in BEHR v3.0B

v3.0B implemented six main changes from v3.0A:

1. Retrievals using daily WRF-Chem profiles use the profile nearest in time to OMI overpass, rather than the last profile before the OMI overpass

2. Ocean surface reflectance calculated at 460 nm instead of 430 nm

3. Variable tropopause pressure (derived from WRF simulations) implemented in the AMF calculation

4. The method for calculating surface pressure from Zhou et al. (2009) was implemented

5. Clear and cloudy scattering weights are included separately in the native pixel files

6. The summary bits in the BEHRQualityFlags field were corrected.

Changes #1–4 directly affect the retrieved VCDs. #5 is intended for advanced users that wish to implement custom profiles. #6 makes rejecting low quality data easier for standard users.





### 4.1 Profile time and ocean LUT effects

Figures 1g and 4d show the changes to the $NO_2$ VCD caused by the change to the wavelength of the ocean reflectance LUT and the selection of the closest profile in time. Figure 1g only shows the effect due to the ocean LUT, as the monthly a priori profiles are not affected by the change in how the closest daily profile in time is selected.

In v3.0A, the ocean surface reflectance was calculated at 430 nm as the approximate midpoint of the wavelength fitting window for an $NO_2$ retrieval (402–465 nm, Krotkov et al., 2017). In v3.0B, this was changed to be 460 nm, which is more consistent with the MODIS band used (459–479 nm). The change in VCD retrieved over ocean is very small ($< 1\%$, Tables 2, 3), as expected.

In v3.0A, when using daily profiles, the last set of profiles before the OMI overpass time was used. In v3.0B, this was

changed to be the nearest profile in time. The overall average is near 0 ($-0.01 \pm 4.6\%$), and the absolute magnitude of the average changes is $< 4 \times 10^{14}$ molec. $cm^{-2}$. As expected, a difference of 1 hour some of the selected profiles makes very little difference to the average retrieved column density.

### 4.2 Implementation of variable tropopause height

BEHR v3.0B uses variable tropopause pressure derived from WRF simulations while in prior versions tropopause pressure is

set to be 200 hPa. Figures 1h and 4e reflect the effect of changes in tropopause pressure on $NO_2$ VCD. The changes in $NO_2$ are consistent with the variation in tropopause pressure. In summertime, the WRF-derived thermal tropopause pressure in lower latitude ($< 45°$ N) is less than 200 hPa. This increases the contribution of the UT, with high sensitivity, to the AMF, which in turn reduces the retrieved $NO_2$ VCDs. In higher latitude ($> 45°$ N), the thermal tropopause pressure is greater than 200 hPa and leads to a slight increase in $NO_2$ VCD. The changes in average $NO_2$ VCD caused by changes in tropopause pressure are

small, $-1.6 \pm 5.3\%$ using monthly average profiles and $-1.1 \pm 8.2\%$ using daily profiles. In wintertime, the WRF tropopause is below the previous 200 hPa value over most of the US and it causes a broad enhancement of $NO_2$ VCD in most US domain ($> 30$ °N) by approximately 2% (Figs. S9, S12, Tables 2, 3).

### 4.3 Surface pressure calculation

Figures 1i and 4f show the impact of switching from a fixed scale height calculation to using the hypsometric equation to

adjust WRF modeled surface pressure to the GLOBE terrain elevation. As expected, the changes are similar whether monthly or daily WRF output is used and are greatest over the Rocky and Appalachian mountains (up to a maximum of $\sim 10\%$). This is similar to the 5% effect Zhou et al. (2009) found in the summer, indicating that the meteorological surface pressure correction in mountainous regions even with a high resolution terrain elevation database.

### 4.4 Publishing separate clear and cloudy scattering weights

In v3.0A and prior versions, an array of scattering weights used in the retrieval was included in the published files in order to allow advanced users to recalculate an AMF using their own a priori $NO_2$ profiles but retain the scattering weights computed




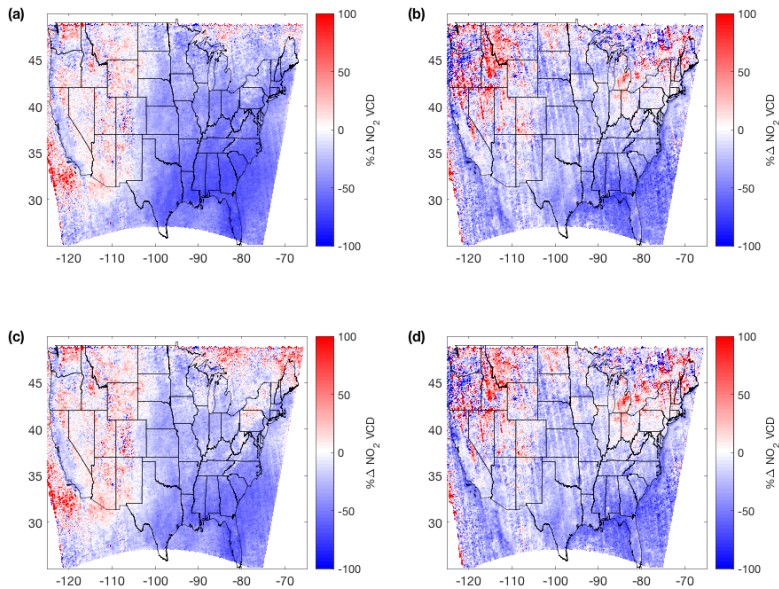

**Figure 7.** Overall average differences in total tropospheric $NO_2$ VCDs between v2.1C and v3.0B for Jun–Aug (a,c) and Jan, Feb, Dec (b,d) of 2012. (a,b) using monthly $NO_2$ profiles in v3.0B, (c,d) using daily profiles in v3.0B.

using the high resolution surface reflectance and elevation data. These scattering weights were the cloud radiance fraction weighted average of the temperature-corrected clear and cloudy scattering weights (Eqs. 2, 6). Using these scattering weights along with the published a priori profiles, users could reproduce BEHR AMFs well, to within $0.5 \pm 1.9\%$. However, publishing the clear and cloudy weights separately increases the precision of reproduced AMFs by three orders of magnitude. Using these with the provided BEHR a priori profiles allows users to reproduce BEHR AMFs effectively exactly (Fig. S7). This also permits users to use different cloud fractions in their custom AMF calculations.

### 4.5 BEHR Quality Flags

Starting with v3.0A, the BEHRQualityFlags field summarized key quality issues from both the NASA and BEHR processing steps. The first and second bits in these values are summary bits, so that users who want high quality data can very easily identify such data. Due to a bug in v3.0A, these bits did not filter out all low quality data. This has been rectified in v3.0B. See Sect. 6 for the proper use of this flags.



## 5 Overall difference

Overall, the two changes that had the largest impact on the retrieved VCDs were the new NASA slant column fitting and the new a priori $NO_2$ profiles ($-14 \pm 14\%$ and $0.86 \pm 20.14\%$, respectively, Table 2). Although the overall average effect of the new profiles is small, this is only because it causes both positive and negative changes to the VCDs. The large standard
deviation reflects how different areas do have very significant changes. The effect of the a priori profiles was especially strong in the SE US where lightning has a strong influence on the profile shape in the summer (Fig. 7). Given the high sensitivity of $NO_2$ retrievals to upper tropospheric $NO_2$, this is not surprising. The omission of lightning $NO_2$ from the original BEHR product was a limitation of WRF-Chem at the time the product was created (Russell et al., 2012); lightning $NO_x$ emission was not added to WRF-Chem until v3.5.0, released in April, 2013 (http://www2.mmm.ucar.edu/wrf/users/download/get_sources.
html#WRF-Chem). The change due to the SCD fitting resulted in a fairly uniform decrease in $NO_2$ VCDs across the domain.

The difference in the averages using daily (Fig. 7c,d) vs. monthly profiles (Fig. 7a,b) is not large, as noted in Laughner et al. (2016), because averaging over time periods greater than a month eliminates the temporal variability captured by the daily profiles. The effect of the daily profiles is on the average strongest in the SE US, as discussed in Sect. 3.4.2, and is still an overall decrease compared to the v2.1C profiles, due to the inclusion of lightning and the reduction in surface emissions.
It should be noted that the difference between retrievals with daily and monthly profiles will be greater in years other than 2012, since the daily profiles incorporate year-specific emissions, while monthly profiles always assume 2012 anthropogenic emissions.

## 6 Recommendations for use

For most users, the quantity of interest will be the standard total tropospheric column contained in the file variable BEHRColum-
nAmountNO2Trop. In order to obtain high quality data, in v3.0B and later, only use pixels or grid cells for which BEHRQual-ityFlags is an even number (i.e. the quality summary bit is 0). This will automatically remove pixels affected by the row anomaly, with cloud fraction $> 0.2$, with low quality surface reflectance, or for which an error occurred during processing. (The quality flags in v3.0A do not properly remove all pixels meeting these criteria.)

Users are encouraged to use years with daily profiles if possible for their application, for two reasons. First, Laughner et al.
(2016) showed that using daily profiles significantly changes day-to-day VCDs, and that some applications of satellite data can be biased when monthly profiles are used. Second, the daily profiles also use year specific emissions (Sect. 2.6.1, so will better capture trends in VCDs as the surface contribution to the a priori profiles is reduced.

For users using BEHR data to evaluate trends, mixing daily and monthly profile retrievals is *not* recommended, as systematic differences between them (i.e. Sect. 3.4.2 of this paper; Laughner et al., 2016) will bias any trends observed. Second, caution
is advised if comparing 2005 or 2006 data using daily profiles to other years; the different WRF-Chem boundary conditions (Sect. 2.6.1) may also bias observed trends.



## 7 Conclusions

Here we present v3.0 of the Berkeley High Resolution OMI NO$_2$ product (BEHR NO$_2$). This version incorporates a number of changes, including updated a priori NO$_2$ profiles with lightning NO$_x$ emissions, daily NO$_2$ profiles for select years, a directional surface reflectance product, variable tropopause height, a new gridding algorithm, and improved surface pressure

calculation, in addition to using the current NASA OMI NO$_2$ Standard Product. The new a priori profiles and the upgrade to the new NASA product had the largest effect on the retrieved total tropospheric VCDs. Retrieved visible-only tropospheric VCDs were most strongly affected by the new visible-only AMF formulation, but otherwise were similarly affected by each change.

## 8 Code and data availability

BEHR data is stored in monthly compressed files as four subproducts on the University of California DASH archive (Laughner et al., 2018a, b, c, d). All BEHR data is also available for download as individual files at behr.cchem.berkeley.edu. The BEHR code is hosted on GitHub at https://github.com/CohenBerkeleyLab/BEHR-core/tree/master (Laughner and Zhu, 2018b). WRF-Chem simulations for 2005, 2007–09, and 2012–14 are available at the time of writing; due to the large file size, access can be arranged by contacting the corresponding author. The analysis code (and its dependencies) along with the incremental averages

are available at https://doi.org/10.5281/zenodo.1247564 (Laughner and Zhu, 2018a).

The v3.0 NASA Aura OMI NO$_2$ standard product (Krotkov and Veefkind, 2016) and OMI/Aura Ground Pixel Corners product (Kurosu and Celarier, 2010) was obtained from the Goddard Earth Science Data and Information Services Center (GES DISC) in Greenbelt, MD, USA. The MODIS Aqua Clouds 5-Min L2 Swath 1 and 5 km (MYD06_L2 Platnick et al., 2015) and MODIS Terra+Aqua BRDF/Albedo Parameters 1–3 Band3 and QA BRDF Quality Daily L3 Global 30ArcSec

CMG V006 (Schaaf, 2015a, b, c, d, MCD43D07, MCD43D08, MCD43D09, MCD43D31) were acquired from the Level-1 and Atmospheric Archive and Distribution System (LAADS) Distributed Active Archive Center (DAAC) located in the Goddard Space Flight Center in Greenbelt, MD (https://ladsweb.nascom.nasa.gov/).

## Appendix A: Published Format

### A1 File structure

BEHR data is published as HDF version 5 files. Each file contains a single, top-level group "Data", which in turn contains each orbit as a child group named "SwathX" where X is the orbit number. The datasets for each orbit are contained in the "SwathX" groups.

Separate HDF files contain data at the native OMI pixel resolution and regridded to $0.05° \times 0.05°$ resolution. The regridded files only contain a subset of the variables stored in the native pixel files. The regridded files contain each orbit gridded

separately; each orbit's grid covers the entire domain retrieved. Grid cells outside each orbit's observed swath contain fill

values. Users can identify whether a file contains gridded information by the dataset level attribute "gridding_method", if present, the file is a gridded file; if absent, the file is a native pixel file. Additionally, the "Description" attribute contained in each swath indicates whether the data is at native or regridded resolution.

Retrievals using daily vs. monthly $NO_2$ a priori profiles are available separately. Retrievals using monthly profiles will be updated as new OMI and MODIS data becomes available. Retrievals using daily profiles are limited by the need to model said profiles; these will become available as modeled $NO_2$ profiles are simulated.

BEHR files are named with the format "OMI_BEHR-**profile_region_version_yyyymmdd**.hdf", where:

– **profile** will be DAILY or MONTHLY, indicating whether daily or monthly $NO_2$ a priori profiles were used

– **region** region retrieved, currently, US = continental United States.

– **version** is the version string (Sect. A3).

– **yyyymmdd** is the date of the observation

This information is also contained as swath level attributes "BEHRProfileMode", "BEHRRegion", "Version", and "Date", respectively.

## A2   Quality flagging

BEHR data contains a 32-bit unsigned integer quality flag field that summarizes quality errors and warnings from both the NASA processing and BEHR processing. Each bit in the integer value represents a specific error or warning flagged during processing. The bits are divided into three categories; the bit number is the position of the bit (1-based) starting from the least significant bit.

– **Bits 1 & 2: summary bits.** These summarize the other 30 bits. Users interested in simple filtering can focus only on these.

– **Bits 3–16: error bits.** These are set to 1 for significant errors in the retrieval that preclude the use of the corresponding $NO_2$ data in any capacity.

– **Bits 17–32: warning bits.** These are set to 1 as non-fatal warnings about the processing of the corresponding data. These do not automatically preclude the use of the corresponding data, but rather provide warnings of potentially lower-quality data or information about decisions made during the retrieval. The flags for low quality BRDF data (Sect. 2.2) fall into this category.

The meaning of each used bit is given in the "FlagMeanings" attribute of the BEHRQualityFlags dataset; here, we will only discuss the two summary bits.

Bit 2 is the error summary bit; it is set to 1 if any error bit is set. Therefore, $NO_2$ columns from any pixel with this bit set should not be used. In v3.0B, this is set if the NASA VcdQualityFlags or XTrackQualityFlags fields indicate the pixel should not be used, or if the BEHR AMF is invalid (usually because a WRF profile is not available for that pixel).

Bit 1 is the quality summary bit; in v3.0B, it is set to 1 if bit 2 is set, the MODIS BRDF coefficients are low quality, or
the OMI geometric cloud fraction exceeds 20%. Therefore, the $NO_2$ data can be restricted to high quality, total tropospheric column data by using only pixels where this bit is not set.

These quality flags focus on the quality of the $NO_2$ retrieval; therefore ancillary data (such as the MODIS surface reflectance or MODIS clouds) is not necessarily unusable for pixels flagged with a retrieval error.

In the gridded product, the quality flags field is a bitwise OR of all contributing pixels' quality flags. Therefore, any error or
warning in a pixel that contributes to a grid cell is propagated to the grid cell.

## A3   Versioning

BEHR versions follow the format "vX-XYrevZ", e.g. v3-0Arev0. The "X-X" indicates the version of the NASA Standard Product that was ingested as the basis for that BEHR retrieval. "Y" is a sequential letter (A, B, C, etc.) indicating the major version of BEHR produced from the same NASA SP base; i.e., v3-0A indicates the first major BEHR version based on the
NASA SPv3. "revZ" (short for "revision") indicates a small update to the BEHR product. Revisions are reserved for small changes that are not expected to significantly affect scientific results obtained from the data; e.g. updates to file format or attributes, or very uncommon error corrections. A revision of 0 may be omitted from the version string, i.e. "v3-0A" and "v3-0Arev0" are the same version.

## A4   Traceability

To ensure traceability, files ingested during processing from other satellite products or models are recorded in the swath level attributes "OMNO2File" (NASA $NO_2$ SP data), "OMPIXCORFile" (pixel corner data), "MODISCloudFiles" (MYD06 files that MODIS cloud data is taken from), "MODISAlbedoFile" (MCD43Dxx files that BRDF parameters are taken from), and "BEHRWRFFile" (WRF-Chem output files the $NO_2$ profiles are taken from; are post-processed for monthly average profiles).

The BEHR code is available on GitHub at https://github.com/CohenBerkeleyLab/BEHR-core (Laughner and Zhu, 2018b).
Each release will be tagged with the same version string as the data. Additionally, eleven swath level attributes contain the Git SHA-1 hash of the most recent commit of the core BEHR code and additional dependencies at the time each of the three major steps in processing BEHR data. These attribute names have the form "GitHead_**repo_step**", where **repo** will be one of:

– **Core**: the core BEHR repository (https://github.com/CohenBerkeleyLab/BEHR-core)

– **BEHRUtils**: the repository of BEHR satellite utility functions (https://github.com/CohenBerkeleyLab/BEHR-core-utils)

– **GenUtils**: the repositiory of general Matlab utilities (https://github.com/CohenBerkeleyLab/Matlab-Gen-Utils)



- **PSM**: the repository containing the modified "omi" python package used for gridding (https://github.com/CohenBerkeleyLab/BEHR-PSM-Gridding)

- **MatPyInt**: the Matlab-Python type conversion interface (https://github.com/CohenBerkeleyLab/MatlabPythonInterface)

- **WRFUtils**: the repository containing Matlab utilities for working with WRF data.

and **step** will be one of:

- **Read**: step in which OMI, MODIS, and GLOBE data are ingested into Matlab and (where necessary) averaged to OMI pixels.

- **Main**: step in which scattering weights and $NO_2$ profiles are matched to OMI pixels, the BEHR AMFs and VCDs are calculated, and the data is gridded.

- **Pub**: step in which the BEHR Matlab files are converted to HDF files.

*Competing interests.* The authors declare no competing interests.

*Acknowledgements.* The authors gratefully acknowledge support from the NASA ESS Fellowship NNX14AK89H, NASA grant NNX15AE37G, and the TEMPO project grant SV3-83019.

We would like to acknowledge high-performance computing support from Cheyenne (doi:10.5065/D6RX99HX) provided by NCAR's
Computational and Information Systems Laboratory, sponsored by the National Science Foundation. This research also used the Savio computational cluster resource provided by the Berkeley Research Computing program at the University of California, Berkeley (supported by the UC Berkeley Chancellor, Vice Chancellor for Research, and Chief Information Officer).

We acknowledge use of the WRF-Chem preprocessor tools MOZBC, fire_emiss, etc. provided by the Atmospheric Chemistry Observations and Modeling (ACOM) laboratory of NCAR. This We also thank Eric Bucsela and Jim Gleason for very helpful discussions about the
new formulation of the visible-only AMF.



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
