# Peer review of "The Berkeley High Resolution Tropospheric NO2 Product"

_Earth System Science Data, 2018_

## Referee Comment (RC1) · Anonymous Referee #1 · 23 Jul 2018

General Comments

This article summarizes updates to the Berkeley OMI NO2 product (BEHR). The authors describe the production of the product and the updates made to their previous version v2.1C and now included in version v3.0A and v3.0B. Overall I believe the quality of the data is acceptable and the analysis is thorough. The BEHR product has been used in several scientific studies and an updated version with the latest model profiles, surface reflectance etc will be welcome. I downloaded some data without any problems, and what I have looked at seems to be very complete.

Many of my critiques are stylistic or requests for clarity. I found this paper quite difficult to get through, despite good organization and good English. There are a lot of details (sometimes, maybe too many), but occasionally something quite key will be missing.

[Figure]

The paper also includes a 16 page supplement (several interesting figures are contained here). A short sentence on one small point may reference two figures in the supplement and two tables within the paper, sending the reader off to more details, and at times I felt a bit confused trying to follow all the references to other sections and the supplement. There are several figures with multiple panels and a lot of detail within the captions, but I think the figures could do with in-figure titles (for instance, add "JJA" and "DJF" to the ones sorted by season). Individual panels are frequently referenced throughout the text, and it's a hard slog to re-read long captions at each reference. Figures are often not introduced in a general way, but an individual panel may be referred to quickly in a discussion. I think the paper would be a lot clearer if each figure were introduced at its first mention in a concise way by something like "Figure 6 shows fill in blank".

The paper describes updates to V3.0A and V3.0B but it wasn't clear to me why there are two versions to discuss. Why not just discuss version 3.0B (presumably the most recent version)? Have people already published with V3.0A? Also, it is not mentioned in the paper what version is provided in the data files.

Specific comments

The title: This paper describes an algorithm update to the product for one specific satellite instrument (OMI). I think it would be more useful to give a more descriptive title, as there already exists a former "Berkeley High Resolution Tropospheric NO2 Product" and perhaps there will be more in the future. Something like: "The Berkeley High Resolution Tropospheric NO2 Product: Updated version 3.0 for the Ozone Monitoring Instrument" for example.

Abstract: Does this paper and dataset only include data over the US? OMI is global but the paper only shows US region so tell the reader early on.

It's not clear to me what years of data are available. You use 2012 as an example throughout the paper but don't talk about other years. I only get that other years other

than 2012 are available because it's mentioned to use some years with caution. Be specific in abstract and text.

You discuss V3.0 here but V3.0A and 3.0B in text. Is this V3.0B that is actually listed and discussed in abstract, and provided on repository?

Page 1, Line 19: While I guess anything in the atmosphere could have some effect on the radiative balance, the radiative effects of NOx of any significance are indirect, not direct.

Page 1, Line 23: You give other thorough details, but then end with "NOx itself is harmful". Be more precise – not clear what this means. Harmful in what way?

Page 2, Line 4: actually, GOME-2 came after OMI. Reword to make sense. For context, give approx spatial resolutions of various resolutions to clarify OMI's ability to look at urban and point sources.

Page 2, Line 17: "also" and "as well" are repetitive

Page 2, Line 24: Lots of molecules are detected using remote sensing in the UV. If you want to mention NO, give specific reason it is not detectable.

Page 3, Line 8: It's not clear to me why the surface elevation is necessary for a radiative transfer model. Is this something in your particular setup?

Page 3, Line 15: You discuss the effects of increasing the resolution in different studies, but relative to what original resolutions?

Page 3, Line 21: "restricted to a region of the world" is awkward phrasing. Any region is a region. Be more specific.

Page 3, Line 22: I don't think you should use the word "retrieval" here to describe your product. That implies you are doing a retrieval, which is really the spectral to slant column step (performed by NASA).

Page 3, Line 24: Is v2.1C described elsewhere? Reference it.

Section 1 in general: I think you should list the science studies that have used the BEHR product specifically so that you can show importance of this product.

Page 3, Line 27: The use of the word "visible" to describe an AMF here and elsewhere is confusing, but particularly in this section, where you have just discussed the visible (spectral) absorption of NO2. I had no idea what you were talking about until later in the paper.

Page 3, Line 31: What were the old emissions? These update descriptions don't make sense without context.

Page 3, Line 2: vague statement

Page 4, Line 20: is g(p) mixing ratio or partial column? Specify

Page 4, Line 24: Confusing to me what is provided in data until I looked at it.

Page 10: I'm not sure why so much detail is provided on RAA here. Is the definition different from what is standard or included in the OMNO2 files? If not, I would suggest you move it to the supplement.

Line 18: Give reference for this empirical correction

Page 6, Line 6: Useful to know spatial resolution here for MODIS products

Page 6, LIne 28: Mention this is modeled resolution

Table 1: What does "atmospheric profile" mean? (p,T, O3?)

Page 7, Line 1: Why do you need an independent surface pressure of an OMI pixel? If the model resolution was larger than OMI, this would be more obvious. Why can't you use some kind of average of the modeled surface pressures?

Page 7, Line 2: For what is tropopause pressure used? Is it to define the height of the troposphere for equation 2?
Page 8, Line 6: You give a lot of other details. . .. But what is source of OMI geometric and radiance cloud fractions? Is cloud fraction from the O2O2 algorithm cloud pressure? Why is there a MODIS cloud fraction included? I don't see this MODIS data mentioned anywhere else so not sure why it's here. It's from a different satellite which seems like a dodgy cloud product for use in OMI analysis.

Section 2.6: Again, I'm confused about what years are analyzed, what years are modeled with WRF-Chem and why. Why is MOZART not available 2005-2006?

Page 8, Line 28: Is MOZART only used for boundary conditions? This sentence is confusing.

Page 8, Line 31: What is GEOS-Chem resolution?

Page 9, Line 14 and Line 21: what does "when possible" mean? What years are available and what determines this?

Page 9, Line 22: I'm not sure I understand what you are doing here. Is this making a single climatology for the entire month? Why not make 24 1-hour climatologies for each month and take the nearest profile of each observations?

Figure 1: Caption needs to mention what this is relative to, and version (ie., changes between V2.1C and V3.0A or whatever).

Since the VIS AMF is actually a separate AMF product in the datafiles, I find it very confusing to have it included in the changed parameters. Could it be moved to its own figure and discussion? It feels confusing to have it brought up in the middle of the other discussions of input parameters.

Page 10, Line 7: Here and elsewhere: Figure and table referred to without contextual introduction. I think Figure 1 should be introduced and discussed concisely before a panel is mention in brackets with other tables).

Table 2: Mention data version and change relative to what.

Tables 2 and 3: Probably unnecessary significant figures after decimal place?

Page 13, Line 1: Specify what is the other approach

Page 14, Line 28: Reference to v3.0B but still in V3.0A section.

Figure 4: First reference is to panel c on page 18, but I think need an introduction to general figure for context. Include some titles on figures panels for readability. Also, mention change is relative to what. And what is the version?

Page 15, Line 6: I find this discussion a bit hard to follow, with references to a previously undiscussed figure (I need a sentence to help me interpret it first) and to the supplement. Also, "it" on line 6 refers to what? What is "order of averaging"?

Figure 5: What is data version here?

Figure 6: Need some intro discussion in text. I found I was with a reference to this figure but not much assistance on how to interpret it.

Page 8, Line 15: Not sure what is (2) at end of sentence.

Section 4.1: You give 6 points early. Break up this section for better organization.

Page 19, Line 10: I believe you mean "overage average difference" instead

Page 19, Line 30: Just to clarify, these scattering weights are from the OMI SCD product?

Page 20, Line 3: I'm not really clear why users could expect to reproduce at all without cloudy scattering weights and surprised it's so small, unless really low cloud fractions are being considered. Seems like this should not be attempted as obviously information is missing. I'm not sure you really need to go into the details here. Just saw you need to provide it for reproducibility/completeness. Line 5: If people start using their own cloud fractions, will they be consistent with the cloud top heights? Not sure this is something that should be encouraged for any except most advanced users.

Page 22, Line 13: Why are some years available and others not? (Can mention earlier in paper.)

Page 25, Line 19: Remove "This"

Technical Comments

Page 1, Line 7: remove "has"

Page 2, Line 34: change "conversion of factor" to "conversion factor"

Page 5, Line 1: Change semicolon to comma

Page 5, Line 2: Change to "denominator in"

Page 6, Line 3: Remove extra "in"

Page 8, Line 20: change to "year's"

Page 10, Line 10: change to "changes in the surface reflectance"

Page 15, Line 8: change to "influenced"

Page 19, Line 11: change to "1 hour in some"

---

## Referee Comment (RC2) · Anonymous Referee #2 · 14 Aug 2018

Improvement to the tropospheric NO2 retrieval is important especially for the widely used instruments like OMI. The authors present in detail the improved BEHR version 3.0 algorithm, concentrating on the most important parameters in NO2 retrieval, e.g., surface albedo, surface pressure, and a priori NO2 profile shape, which is potentially beneficial for studies focusing on high resolution application. The dataset is accessible to the public and is consistent with the description in this manuscript. However, I have got the feeling, at least from the abstract, that the manuscript is more suitable to journals like AMT or ACP, since it is mainly talking about algorithm instead of the dataset itself. Therefore, I would suggest the authors to update the abstract and maybe also the conclusion with more descriptions about the dataset and revise the manuscript addressing the following comments:

general: Despite the good written language, the organization sometimes makes it difficult for me to identify which improvement is for v3.0A which is for v3.0B. For instance, in the "Methods" section, some methods are introduced for v3.0A and some are for v3.0B. Also, the methods of older version are sometimes introduced in "Methods" section (e.g. surface pressure) and sometimes in each subsection (e.g. visible-only AMF calculation). Therefore, this methods section is not fully referable when reading the following sections. In addition, illustrations like "figure 1 shows..." and "table 1 shows..." are missing.

specific:

page2line4 GOME2 (GOME2A in 2006 and GOME2B in 2012) is newer than OMI.

page2line24 I believe the difficulty of NO observation is not only because of the absorption in UV. Even it is noisier than VIS, it still works for gases, e.g. O3 and even possible for NO2. Please specify this sentence.

page2line25 "inferences about total NOx are made from NO2 measurements" maybe also because of the quick conversion of NO to NO2?

page3line4 I suggest including the TM5 also as examples, since it is largely used currently, e.g. for OMI and TROPOMI retrieval.

page3line3-11 I recommend combining these two paragraphs together, since they both talk about how to calculate AMFs, and the "input data" in line 12 talks mainly about the input data (i.e. profiles) in the 1st paragraph.

page5line8 The definition of RRA it not that special, can be removed.

page6line10 What does "BEHR uses the file dated for the day being retrieved for the BRF coefficients." mean?

page8line7 Why does the BEHR include these different cloud products? For instance, is there a specific reason to include MODIS cloud fraction? Please also add more

information about the OMI-derived quantities. Do you retrieve the cloud fractions, or do you use OMCLDO2 or OMCLDRR? Also, please update the expression "radiance cloud fraction" to cloud radiance fraction here and through the manuscript.

page8line28 What output is used here?

page9line7 When would this extrapolation happen? If it is because of the different surface pressure from scattering weight and profiles, then it might be even better to shift the profile but not extrapolate.

page9line14 Please specify "when possible".

page11table2 What is ocean LUT here and through the manuscript? Do you mean ocean reflectance LUT?

page11figure2 This figure is comparing with BEHR v3.0B but it is described in the "Changes in BEHR v3.0A" text. Additionally, the interpretation after figure is talking about the changes in surface reflectance over land. Please add more analysis about changes over water, since the difference is quite significant.

page13figure3 The figures are not clear and the conclusions are not convincing to me. Why is there a straight line (a deep blue line with no percent change in NO2 near cloud pressure 700 hPa in (a) and a distinct line near the black dashed line in (b))? What is the definition of surface NO2 concentration and why does it matter here? Since in Eq. 2, it is only the profile shape (relative vertical distribution) matters in the AMF calculation but not the absolute concentration. Also, I am not sure with "Greater percent difference with greater surface NO2 concentration.", because most of the largest differences in the figure are found for low surface NO2 concentration (blue to deep blue in (a)). Similarly, greater percent differences are also found for small difference between the cloud fraction and cloud radiance fraction, since quite a lot of yellow dots are close to the black dashed line in (b).

page14line21 What is UT?

page15figure4 the (a) and (b) panels are not described anywhere.

page15line3 I do not understand this hypothesis, since the profile shape itself depends not on the cloud information. The selection criteria of cloud fraction 0.2 only has impact on the NO2 column calculation, and it has no impact on the profile shape.

page17line5 Is there probably a name or reference to this temperature profile?

page18line6-13 There are introductions of the previous method and why PSM method cannot be used here. It might be better to add a small introduction of the new CVM which is actually used in this study.

page19line7 The ocean reflectance is calculated without MODIS data, therefore I do not understand the goal of this change to 460 nm. Even the impact is small, the reflectance at 430 nm shall be used because of the reason exactly described in the text.

technical:

page7line2 change "within IN" to THE

page8line20 add "that year'S total"

page14line15 add "meteorology ARE also"

page15line6 remove "it IS seems"

page15line8 change "INFLUENCED by lighting"

page18line11 change "algorithm RESULTS"

page18line15 remove (2)

---

## Author Comment (AC1) · 12 Sep 2018

**The Berkeley High Resolution Tropospheric NO$_2$ Product**
**Response to Anonymous Referee #1**

Joshua L. Laughner, Qindan Zhu, and Ronald C. Cohen

September 11, 2018

We appreciate the reviewers careful reading and suggestions to improve the readability of the paper. We realize that there is a great deal of information here and that organizing better will to minimize the strain on the reader. We believe that incorporating many of the reviewer's suggestions help in this matter.

Regarding the overall comment about the complexity of the paper's organization, we have taken this to heart and tried to address these concerns throughout (adding titles to figure panels, introducing entire figures where appropriate, and moving some figures from the supplement into the main paper). The most significant change we made was to add a new section, "Paper structure", after the methods and before the results. Here we outline the remainder of the paper, directing readers to the relevant sections if they are more interested in the changes to the algorithm, usage recommendations, or product description, and introducing the two main figures and tables that most of the discussion refers to.

Responses to specific comments follow. The reviewer's comments will be shown in red, our response in blue, and changes made to the paper are shown in black block quotes. Unless otherwise indicated, page and line numbers correspond to the original paper. Figures, tables, or equations referenced as "R$n$" are numbered within this response; if these are used in the changes to the paper, they will be replaced with the proper number in the final paper. Figures, tables, and equations numbered normally refer to the numbers in the original discussion paper.

The paper describes updates to V3.0A and V3.0B but it wasn't clear to me why there are two versions to discuss. Why not just discuss version 3.0B (presumably the most recent version)? Have people already published with V3.0A? Also, it is not mentioned in the paper what version is provided in the data files

v3.0A was available on our website for a 9-month period between November 2017 and July 2018. Just in case anyone downloaded that version and publishes results with it, we want there to be a record documenting the changes since v3.0A. We have added a paragraph clarifying this to the introduction:

> "v3.0A was available on the BEHR website (`behr.cchem.berkeley.edu`) between November 2017 and July 2018; v3.0B replaced v3.0A on the website and the static repositories (Laughner et al., 2018a,b,c,d) in July 2018. Therefore, in this paper, we will separate changes implemented in v3.0A from those in v3.0B,

so that the differences between v3.0A and v3.0B can be accounted for if any results are published using v3.0A."

The title: This paper describes an algorithm update to the product for one specific satellite instrument (OMI). I think it would be more useful to give a more descriptive title, as there already exists a former "Berkeley High Resolution Tropospheric NO2 Product" and perhaps there will be more in the future. Something like: "The Berkeley High Resolution Tropospheric NO2 Product: Updated version 3.0 for the Ozone Monitoring Instrument" for example.

We appreciate the desire for clarity; however we envision taking advantage of ESSD's living data philosophy to update this paper to describe future versions of the product, so tying the paper title to a specific version is counterproductive, especially as our version numbering system is such that "3.0" indicates the version of the NASA Standard OMI $NO_2$ product ingested as the base, if NASA updates their version, our version will also increment.

Abstract: Does this paper and dataset only include data over the US? OMI is global but the paper only shows US region so tell the reader early on. It's not clear to me what years of data are available. You use 2012 as an example throughout the paper but don't talk about other years. I only get that other years other than 2012 are available because it's mentioned to use some years with caution. Be specific in abstract and text. You discuss V3.0 here but V3.0A and 3.0B in text. Is this V3.0B that is actually listed and discussed in abstract, and provided on repository?

Correct, the data only includes data over the US and parts of Canada. We have clarified the geographic region and time periods covered in the abstract:

> "...BEHR v3.0B builds on the NASA version 3 standard Ozone Monitoring Instrument (OMI) tropospheric $NO_2$ product to provide a high spatial resolution product for a domain covering the continental United States and lower Canada that is consistent...

> ...The subproducts using monthly profiles are available from January 2005 to July 2017. The subproducts using daily profiles are currently available for years 2005–2010 and 2012–2014. 2011 and 2015 on will be added as the necessary input data are simulated for those years."

Further, at the end of the introduction, we also clarify why only certain years are available, and explicitly state that we are using only 2012 here as an example:

> "Because of the computational resources required to simulate daily a priori $NO_2$ profiles, BEHR v3.0B is produced for all years from 2005 on using monthly average $NO_2$ profiles, and for as many years as possible with daily $NO_2$ profiles. The latter is available for 2005–2010 and 2012–2014, with the remaining years following as the simulations of the necessary $NO_2$ profiles are completed. In this paper, we focus on the 2012 data as an example to understand the effect each change to the algorithm has on the final VCDs."

Page 1, Line 19: While I guess anything in the atmosphere could have some effect on the radiative balance, the radiative effects of NOx of any significance are indirect, not direct.

Section 4 of Kiehl and Solomon (1986) indicated that $NO_2$ direct heating of the stratosphere is usually a few percent, but may be more under certain conditions. However, since our focus is tropospheric $NO_2$, we have deleted this sentence.

Page 1, Line 23: You give other thorough details, but then end with "NOx itself is harmful". Be more precise – not clear what this means. Harmful in what way?

We have added:

" Additionally, $NO_x$ itself is harmful, **as, for example, exposure causes bronchoconstriction and associated difficulty breathing, especially for those affected by asthma** (Kagawa, 1985; Chauhan et al., 1998; Wegmann et al., 2005; Kampa and Castanas, 2008)."

Page 2, Line 4: actually, GOME-2 came after OMI. Reword to make sense. For context, give approx spatial resolutions of various resolutions to clarify OMI's ability to look at urban and point sources.

We removed GOME-2 from the "early" instruments and added resolutions:

"The spatial resolution available with early instruments (i.e. the Global Ozone Monitoring Experiment, GOME, $40 \times 320$ km$^2$, Burrows et al. 1999b; the SCanning Imaging Absorption SpectroMeter for Atmospheric CHartographY, SCIA-MACHY, $30 \times 60$ km$^2$, Noel et al. 1998) allowed inferences at the scale of entire continents or entire metropolitan regions, including cities and their surroundings. More recent instruments have much higher resolution (e.g. the Ozone Monitoring Instrument, OMI, $13 \times 24$ km$^2$, Levelt et al. 2006; the Tropospheric Monitoring Instrument, TROPOMI, $7 \times 7$ km$^2$, Veefkind et al. 2012), allowing inferences about individual point sources and urban cores."

Page 2, Line 17: "also" and "as well" are repetitive
Removed "also"

Page 2, Line 24: Lots of molecules are detected using remote sensing in the UV. If you want to mention NO, give specific reason it is not detectable.

As NO measurements are not important for understanding our product, we have modified this to simply state that NO is not measured by the current sensors in orbit:

"The current fleet of space-based sensors measures $NO_2$, not total $NO_x$, but due to the rapid daytime equilibrium between NO and $NO_2$, this allows inferences about tropospheric $NO_x$ to be made from $NO_2$ measurements."

Page 3, Line 8: It's not clear to me why the surface elevation is necessary for a radiative transfer model. Is this something in your particular setup?

[Figure]

Figure R1: Two vectors of scattering weights for solar and viewing zenith angles of 30°, a relative azimuth angle of 180°, and a surface albedo of 0.02.

To our knowledge, all satellite $NO_2$ products need to know the surface elevation (or equivalently surface pressure) in order to get the lower boundary condition for radiative transfer correct. If there was a dark surface at a high altitude, for example, sensitivity would decrease more at higher altitudes because fewer photons would be scattered back to space before reaching the ground, where they would likely be absorbed (see Fig. R1). As this is standard in satellite $NO_2$ retrievals, we elected not to add any text explaining this to the introduction.

Page 3, Line 15: You discuss the effects of increasing the resolution in different studies, but relative to what original resolutions?

We have noted the resolution of the $NO_2$ profiles used in the global retrievals that these papers compare to:

> "Russell et al. (2011) found that increasing the resolution of the $NO_2$ profiles **from $2.5° \times 2°$ to 4 km** altered the retrieved VCDs by up to 75%, primarily by capturing the urban-rural gradient in surface $NO_2$ concentrations. McLinden et al. (2014) found that increasing the a priori profiles' resolution **from $3° \times 2°$** to 15 km resulted in a factor of 2 increase in $NO_2$ column over the Canadian oil sands."

Page 3, Line 21: "restricted to a region of the world" is awkward phrasing. Any region is a region. Be more specific.

We have clarified:

> "The current trade off to obtain such high resolution profiles is that the resulting product **is only available over a subset of the world, rather than globally.**"

Page 3, Line 22: I don't think you should use the word "retrieval" here to describe your product. That implies you are doing a retrieval, which is really the spectral to slant column step (performed by NASA)

Our experience has been that "retrieval" is often used as a shorthand for the entire $NO_2$ algorithm or one of the components (SCD fitting or AMF calculation). Nevertheless, we have changed to "product" here and elsewhere in the paper.

Page 3, Line 24: Is v2.1C described elsewhere? Reference it.
Added:

> "Here we describe the updates from v2.1C to v3.0B. (For information on v2.1C, see Russell et al. 2011 and the changelog at `http://behr.cchem.berkeley.edu/Portals/2/Changelog.txt`.)"

Section 1 in general: I think you should list the science studies that have used the BEHR product specifically so that you can show importance of this product.
We have added the following near the end of the introduction:

> "The Berkeley High Resolution (BEHR) Ozone Monitoring Instrument (OMI) $NO_2$ retrieval is one such regional product that provides tropospheric $NO_2$ VCDs over the continental United States using high resolution a priori inputs. **The BEHR product has been used in numerous studies covering areas of research such as $NO_x$ trends (Russell et al., 2012; Kharol et al., 2015; Pusede et al., 2016; Parker et al., 2017), anthropogenic emissions (de Foy et al., 2015; Jiang et al., 2018), soil emissions (Hudman et al., 2012), land use regression modeling (Bechle et al., 2015), and model evaluation (Canty et al., 2015; Travis et al., 2016).**"

Page 3, Line 27: The use of the word "visible" to describe an AMF here and elsewhere is confusing, but particularly in this section, where you have just discussed the visible (spectral) absorption of NO2. I had no idea what you were talking about until later in the paper.
We see how that could be confusing. Per one of your other suggestions, we moved discussion of the change to the visible-only AMF to the supplement, which should distance it from the discussion of the visible spectral range. Therefore, we removed this point in the itemized list, and added the following paragraph after the list:

> "These changes all affect the tropospheric VCDs. BEHR also provides a "visible-only" VCD, that is, the VCD excluding $NO_2$ below clouds for users interested in e.g. cloud slicing methods (Choi et al., 2014). These visible-only VCDs are computed by dividing the tropospheric slant columns by the corresponding visible-only AMF BEHR v3.0A implemented a more physically intuitive form of the visible-only AMF than that in v2.1C. This change is described in the supplement for interested users."

Page 3, Line 31: What were the old emissions? These update descriptions don't make sense without context.
We have added:

> "Monthly profiles use 2012 emissions, **instead of 2005 emissions used in v2.1C and prior**...Daily profiles, **with year-specific emissions**, used for as many years as possible"

Page 3, Line 2: vague statement
We are unsure which statement this refers to.

Page 4, Line 20: is g(p) mixing ratio or partial column? Specify
Mixing ratio, specified in text

Page 4, Line 24: Confusing to me what is provided in data until I looked at it.
Assuming this comment is directed to the regular VCDs vs the visible-only VCDs, we have specified what the variable names are in the data:

> "This method produces VCDs that include an estimated below cloud component, and thus can be considered a total tropospheric column. This is desirable for applications focusing on near-surface $NO_2$, **and are stored in the BEHR data as "BEHRColumnAmountNO2Trop"**. Other applications...
>
> ...yields a visible-only $NO_2$ column as the output, **stored in the variable "BEHRColumnAmountNO2TropVisOnly" in the BEHR files.** The form of this visible AMF changed from v2.1C to v3.0A; please see Sect. S1 in the Supplement for details of the old calculation."

Page 10: I'm not sure why so much detail is provided on RAA here. Is the definition different from what is standard or included in the OMNO2 files? If not, I would suggest you move it to the supplement.
We have found that different communities use different conventions for RAA; some define 0 as the sun and viewer are on the same side, others define it as the sun and viewer are directly opposite each other. We prefer to be explicit about how we define RAA so there is no confusion, and this seems the logical place to do so.

Line 18: Give reference for this empirical correction
Added:

> "A temperature correction, $\alpha(p)$ **(Bucsela et al., 2006, 2013)**,..."

Page 6, Line 6: Useful to know spatial resolution here for MODIS products
Added:

> "The coefficients, $f_{iso}$, $f_{vol}$, and $f_{geo}$, are taken **at 30 arcsec resolution** from the MODIS MCD43D07..."

Page 6, LIne 28: Mention this is modeled resolution
We have tried to clarify this. This line now reads:

"The ratio of upwelling to downwelling radiation was simulated for 18 solar zenith angles (0° to 85° at 5° increments)."

Table 1: What does "atmospheric profile" mean? (p,T, O3?)
In the table, we have clarified that "MLS" means "mid-latitude summer" and not "Microwave Limb Sounder", in the caption we describe what quantities this concerns:

""Atmospheric profile" refers to the distribution of total precipitable water, $O_3$, $CO_2$, and $CH_4$."

Page 7, Line 2: For what is tropopause pressure used? Is it to define the height of the troposphere for equation 2?
We have added the following to the beginning of Sect. 2.4:

"**For the upper integration limit in Eq. (2)**, BEHR v3.0A and prior versions used a fixed tropopause pressure"

Page 8, Line 6: You give a lot of other details. But what is source of OMI geometric and radiance cloud fractions? Is cloud fraction from the O2O2 algorithm cloud pres- sure? Why is there a MODIS cloud fraction included? I don't see this MODIS data mentioned anywhere else so not sure why it's here. It's from a different satellite which seems like a dodgy cloud product for use in OMI analysis.
We had specified that the OMI cloud products are the same as those in the NASA standard product. We have expanded this to identify them directly:

"BEHR contains several cloud fraction products: **a geometric cloud fraction derived from the $O_2$-$O_2$ algorithm (Acarreta et al., 2004), a cloud radiance fraction calculated by NASA from the $O_2$-$O_2$ product,** and a geometric cloud fraction derived from the Aqua MODIS instrument (which currently makes observations $\sim$ 8 min before OMI), and cloud pressure from the OMI $O_2$-$O_2$ algorithm (Acarreta et al., 2004)."

The MODIS cloud fraction is included because Russell et al. (2011) found it to be less succeptible to errors caused by high surface reflectivity. We retain it in the product as an alternate method of cloud filter for users running into similar issues. The Aqua satellite precedes Aura by about 8 minutes currently, so they are reasonably coincident in time. However, for the AMF calculation we use the OMI cloud fractions, because the MODIS cloud data is not available for every pixel, and because we need a cloud radiance fraction, not just a geometric fraction, for the AMF calculation. We have added text explaining this:

"Russell et al. (2011) found that the MODIS cloud product was less likely to give erroneously large cloud fractions due to high surface reflectivity over the California and Nevada desert, and concluded that this more than offset any error caused by the small separation between the overpass times (currently $\sim 8$ min) of OMI on board the Aura satellite and MODIS on board the Aqua satellite. We continue to provide the MODIS cloud product for cloud filtering; however, because it does not cover the full OMI swath, we use the OMI cloud fractions in the AMF calculations."

Section 2.6: Again, I'm confused about what years are analyzed, what years are modeled with WRF-Chem and why. Why is MOZART not available 2005-2006?

We added a paragraph in response to your earlier comment about what years are analyzed in this paper vs. available in the BEHR product. We have also added the following text at the beginning of section 2.6 that details why there are different sets of a priori profiles:

"From v3.0A onward, BEHR is divided into two subproducts which differ in the temporal resolution of the a priori $NO_2$ profiles. Based on the results in Laughner et al. (2016), using a priori profiles specifically simulated for each day of BEHR observations is preferable; however, the computational cost of doing so limits the time periods that such profiles can be simulated for. Therefore a second subproduct using monthly average profiles derived from the 2012 a priori profiles is available that covers all years of the OMI data record. This assumes that monthly average profiles are applicable to years other than that for which they were simulated; while not a perfect assumption, it has successfully been used in previous $NO_2$ products (e.g. Bucsela et al., 2013).

In this section, we describe the model configuration used to generate the a priori profiles. General model settings will be described first, followed by information specific to the implementation of daily and monthly average profiles in the BEHR algorithm."

As to why MOZART is not available, we are not running MOZART, NCAR provides MOZART output from 2007 on to give chemical boundary conditions for WRF-Chem. Why 2007 is the starting point isn't clear, but since we do not have experience with MOZART and *do* have experience with GEOS-Chem, we elected to use the latter for the two years of the OMI record that MOZART data isn't available for.

Page 8, Line 28: Is MOZART only used for boundary conditions? This sentence is confusing.
Yes, we have clarified:

"**Chemical boundary conditions for WRF-Chem are taken two different global models.** For model years 2007 and later, **chemical concentrations** from the Model for Ozone and Related chemical Tracers"

Page 8, Line 31: What is GEOS-Chem resolution?
Added:

"...the chemical data is taken from the GEOS-Chem model v9-02 (at $2.5° \times 2.0°$ resolution), with updates from... "

Page 9, Line 14 and Line 21: what does "when possible" mean? What years are available and what determines this?
Computational cost is the limiting factor as each year costs about 300,000 core-hours to simulate. We have clarified:

"We make use of daily profiles for **as much of the OMI data record as it is computationally feasible to simulate these profiles**....

As of this writing, daily profiles have been simulated for 2005 to 2010 and 2012 to 2014. Profiles for 2011 are in progress, and profiles for 2015 and later years will be simulated as time and computational resources permit....

Given the computational cost in producing daily a priori profiles, we continue to use monthly average profiles as well to cover years **for which daily a priori profiles have not yet been simulated.**"

Page 9, Line 22: I'm not sure I understand what you are doing here. Is this making a single climatology for the entire month? Why not make 24 1-hour climatologies for each month and take the nearest profile of each observations?
Correct, this creates a single climatology for the entire month. Your suggestion is also a good way to do that, but not one we considered when designing the algorithm. This method of creating a single monthly climatology is in line with the heritage method used in earlier versions of BEHR.

Figure 1: Caption needs to mention what this is relative to, and version (ie., changes between V2.1C and V3.0A or whatever).
As mentioned at the beginning, we added a new section on the paper structure that describes this:

"In sections 4 and 5, we evaluate the effect each change to the BEHR algorithm between v2.1C and v3.0B had on the tropospheric VCDs. In order to provide a clear history, changes introduced in v3.0A will be discussed first (Sect. 4), followed by changes introduced in v3.0B (Sect. 5). V3.0A incorporated all changes up through the introduction of the new gridding algorithm; the remainder are added in v3.0B. Changes to the visible-only VCDs (i.e. those excluding the below-cloud column) are discussed in the supplement (Sect. S1). Following this the overall difference between v2.1C and v3.0B will be presented in Sect. 6. Recommendations for the use of the product are given in Sect. 7. A description of the data format is given in Appendix A.

For the discussion of how changes to the algorithm affect the $NO_2$ VCDs, figures 1 and 2 and Tables 3 and 4 are the central focus. Each panel shows the change in the BEHR $NO_2$ VCDs resulting from a specific change to the algorithm. To generate these figures, BEHR VCDs were computed after adding each change to the algorithm incrementally. Each panel in the figures and line in the tables shows the percent change in VCDs due to the corresponding change to the algorithm. These are computed relative to VCDs with one fewer change to the algorithm; for example, Fig. 1b is the percent difference between VCDs using the new NASA SCDs and the new MODIS BRF surface reflectance versus VCDs using just the new NASA SCDs. The (a) panels in Figs. 1 and 2 and the first lines in Tables 3 and 4 are relative to BEHR v2.1C.

Figure 1 shows the percent change of average BEHR tropospheric VCDs due to each algorithm improvement for the subproduct using monthly average $NO_2$ a priori profiles, while Fig. 2 shows the changes to the subproduct using daily $NO_2$ a priori profiles. (Figure 2 has fewer panels than Fig. 1 as daily profiles were only possible in increments after the change to the algorithm to introduce the new a priori profiles was implemented.) Both figures are for summer (June–Aug.) 2012. Winter changes are presented in the supplement.

Table 3 gives the mean and median changes for each incremental improvement shown in Figs. 1 and 2; that is, it gives the domain-wide mean and median values of the time-averaged changes shown in the figures. Table 4 is similar, but is the statistics for individual pixels, rather than the time-averaged changes."

While this seems complex, it was much more sensible during development to add each change incrementally, rather than implement each change separately then merge them all together.

Since the VIS AMF is actually a separate AMF product in the datafiles, I find it very confusing to have it included in the changed parameters. Could it be moved to its own figure and discussion? It feels confusing to have it brought up in the middle of the other discussions of input parameters.

Good point. We agree and have moved this discussion to the supplement, so that the majority of readers who will be interested in the regular VCD/AMF will not be burdened by it.

Page 10, Line 7: Here and elsewhere: Figure and table referred to without contextual introduction. I think Figure 1 should be introduced and discussed concisely before a panel is mention in brackets with other tables).

As mentioned two comments ago, we've added a section before the existing results sections that introduces Figures 1 and what was Figure 4. Figure 4 has been moved to be Figure 2 so that both can be introduced together, along with Tables 2 and 3.

Table 2: Mention data version and change relative to what.

This is included in the addition from three comments prior.

Tables 2 and 3: Probably unnecessary significant figures after decimal place?

Adjusted so that no number has $> 1$ significant figure after the decimal place (though that significant figure is sometimes in the hundreds) and the $1\sigma$ or quantile value adjusted to end at the same place as the main value. These are expressions of variability, not uncertainty, so it does not make sense to truncate the main value at the first digit of the $1\sigma$ or quantile value.

Page 13, Line 1: Specify what is the other approach

Note: this is now in the supplement. Specified as:

"Although both approaches to calculating a visible-only AMF (i.e. Eq. S1 and Eq. 3) are conceptually valid..."

For reference, Eq. S1 is:

$$A_{\text{BEHR,vis}} = (1 - f)A_{\text{clear,vis}} + fA_{\text{cloudy,vis}}$$

and Eq. 3) is

$$A_{\text{BEHR,vis}} = \frac{(1 - f) \int_{p_{\text{surf}}}^{p_{\text{trop}}} w_{\text{clear}}(p)g(p)\,dp + f \int_{p_{\text{cloud}}}^{p_{\text{trop}}} w_{\text{cloudy}}(p)g(p)\,dp}{(1 - f_g) \int_{p_{\text{surf}}}^{p_{\text{trop}}} g(p)\,dp + f_g \int_{p_{\text{cloud}}}^{p_{\text{trop}}} g(p)\,dp}$$

Page 14, Line 28: Reference to v3.0B but still in V3.0A section.

Corrected.

Figure 4: First reference is to panel c on page 18, but I think need an introduction to general figure for context. Include some titles on figures panels for readability. Also, mention change is relative to what. And what is the version?

This has been addressed in the new "Paper structure" section.

Page 15, Line 6: I find this discussion a bit hard to follow, with references to a previously undiscussed figure (I need a sentence to help me interpret it first) and to the supplement. Also, "it" on line 6 refers to what? What is "order of averaging"?

We have rewritten this section to help with clarity. In the revised version, we do not discuss the hypothesis involving cloud fractions, since it turned out to not be true. The new discussion here focuses on the qualitative difference between monthly and daily profiles, while a mathematical line of reasoning is left to the supplement for readers seeking a deeper understanding. The new section is:

"Figure 4 shows the difference in v3.0A of the average total tropospheric $NO_2$ columns when using daily $NO_2$ profiles rather than monthly average profiles. Figure 4a is the summer (JJA) average, and shows a significant increase in VCDs along the eastern US, which is not present in the winter (DJF) average (Fig. 4b). The timing and location suggests that this difference is due to lightning, as the southeast US especially has very active lightning (Laughner and Cohen, 2017; Travis et al., 2016; Hudman et al., 2007).

Ultimately, the fact that lightning is an intermittent but significant $NO_x$ source in the upper troposphere (UT) is the cause of this difference. Figure 5a shows the statistical distribution of $NO_2$ in the UT for two regions in the US: the southeast, which has significant lightning activity, and the northwest which has very little lightning. The distribution is highly skewed with a long tail in the southeast US due to the lightning activity, but not in the northwest US. Because of the nonlinear nature of the AMF calculation, this skewed distribution translates into different average VCD values.

Figure 5, panels b and c show average shape factors derived from monthly averaged and daily a priori profile for the southeast and northwest US. A shape factor is a profile divided by its integral:

$$S(p) = \frac{g(p)}{\int_{p_{\text{surf}}}^{p_{\text{trop}}} g(p)\, dp}$$

A shape factor can be intepreted as the relative vertical distribution of $NO_2$. It appears implicitly in the AMF calculation (Eq. 2).

Here we see how the skewed UT $NO_2$ distribution affects the southeast US AMFs through the shape factor. Figure 5b shows that the statistically skewed UT $NO_2$ distribution causes shape factors calculated from the monthly average a priori profiles in the southeast US to have a larger fraction of the column $NO_2$ in the UT than that calculated from the daily profiles. Through Eq. (2), this leads to systematically greater AMFs (and therefore smaller VCDs) in the southeast when using the monthly profiles if the scattering weights ($w(p)$ in Eq. 2) are greater in the UT than near the surface, which is usually the case. In contrast, Fig. 5c shows no difference in the monthly or daily shape factors for the northwest US. For interested readers, a more mathematical argument is given in Sect. S2 of the supplement.

The implication is that, for regions with long-tailed statistical distributions of $NO_2$ concentrations, there will be systematic differences between a product using monthly average and daily a priori profiles. It is likely that the VCDs calculated using the daily a priori profiles are more accurate, because in theory daily a priori profiles should properly account for that long tail on days when it is relevant, whereas monthly profiles will average in the extreme values.

Finally we note that this difference between daily and monthly profiles may change in the future. Laughner et al. (2018e) found that the simulation providing the $NO_2$ profiles had too much lightning in the southeast US. Correcting that may reduce the skewness of the UT $NO_2$ distribution. Work is underway to improve the representation of lightning for the southeast US $NO_2$ profiles. "

Figure 5: What is data version here?
v3.0A—added to caption.

Figure 6: Need some intro discussion in text. I found I was with a reference to this figure but not much assistance on how to interpret it.

This figure has been simplified, and in the rewritten section above it is discussed:

> "Figure 5a shows the statistical distribution of $NO_2$ in the UT for two regions in the US: the southeast, which has significant lightning activity, and the northwest which has very little lightning. The distribution is highly skewed with a long tail in the southeast US due to the lightning activity, but not in the northwest US....

> ...Figure 5, panels b and c show average shape factors derived from monthly averaged and daily a priori profile for the southeast and northwest US....Figure 5b shows that the statistically skewed UT $NO_2$ distribution causes shape factors calculated from the monthly average a priori profiles in the southeast US to have a larger fraction of the column $NO_2$ in the UT than that calculated from the daily profiles. Through Eq. (2), this leads to systematically greater AMFs (and therefore smaller VCDs) in the southeast when using the monthly profiles if the scattering weights ($w(p)$ in Eq. 2) are greater in the UT than near the surface, which is usually the case. In contrast, Fig. 5c shows no difference in the monthly or daily shape factors for the northwest US."

Page 8, Line 15: Not sure what is (2) at end of sentence.
Was meant to be a reference to a table—fixed.

Section 4.1: You give 6 points early. Break up this section for better organization.
Done.

Page 19, Line 10: I believe you mean "overage average difference" instead
Overall average difference, yes

Page 19, Line 30: Just to clarify, these scattering weights are from the OMI SCD product?
No these are scattering weights used in BEHR. We have clarified:

> "...an array of scattering weights **used in the BEHR AMF calculation** is included..."

Page 20, Line 3: I'm not really clear why users could expect to reproduce at all without cloudy scattering weights and surprised it's so small, unless really low cloud fractions are being considered. Seems like this should not be attempted as obviously information is missing. I'm not sure you really need to go into the details here. Just saw you need to provide it for reproducibility/completeness. Line 5: If people start using their own cloud fractions, will they be consistent with the cloud top heights? Not sure this is something that should be encouraged for any except most advanced users.

The old scattering weights were not just the clear sky scattering weights, they were, as we said, the weighted average of the clear and cloudy scattering weights, so that users only had to deal with a single vector per pixel. The cloud fractions and pressures were intrinsic to

the published scattering weights, because the cloudy weights were set to 0 below the cloud pressure, and because the published weights were this cloud radiance fraction weighted sum. We have added some equations to clarify this:

"In BEHR v3.0A and prior, these scattering weights were the cloud radiance fraction weighted average of the temperature-corrected clear and cloudy scattering weights:

$$w'(p) = (1 - f)w_{\mathrm{clear}}(p)\alpha(p) + fw_{\mathrm{cloudy}}(p)\alpha(p)$$

where $\alpha(p)$ is defined by Eq. (6) and $w_{\mathrm{clear}}(p)$ and $w_{\mathrm{cloudy}}$ set to 0 below the surface and cloud pressures, respectively.

Using these scattering weights along with the published a priori profiles, users could reproduce BEHR AMFs well, to within $0.5 \pm 1.9\%$, using:

$$A' = \frac{\int_{p_{\mathrm{surf}}}^{p_{\mathrm{trop}}} w'(p)g(p)\, dp}{\int_{p_{\mathrm{surf}}}^{p_{\mathrm{trop}}} g(p)\, dp} \tag{R1}$$

"

Indeed, this is something that only very advanced users should attempt, and the issue of inconsistent cloud fractions and cloud pressure is definitely a concern, but not one that can be rectified easily without a user running the full algorithm, since a different cloud pressure would change the scattering weights by more than just where the cloudy ones are cut off. We have added cautions about this:

"Very advanced users may wish to recalculate custom AMFs using their own $NO_2$ profiles but with the scattering weights used in BEHR....

...The primary purpose is to allow users to replace the BEHR $NO_2$ profiles with their own for a custom AMF calculation. In theory, this also permits advanced users to use different cloud fractions in their custom AMF calculations, but doing so would require careful attention to possible errors, as the scattering weights are tied to the cloud pressure used in BEHR."

Page 22, Line 13: Why are some years available and others not? (Can mention earlier in paper.)
We have addressed this in the intro and methods

**References**

Acarreta, J. R., De Haan, J. F., and Stammes, P.: Cloud pressure retrieval using the O2-O2 absorption band at 477 nm, J. Geophys. Res. Atmos., 109, D05 204, doi:10.1029/2003JD003915, URL http://dx.doi.org/10.1029/2003JD003915, 2004.

Bechle, M. J., Millet, D. B., and Marshall, J. D.: National Spatiotemporal Exposure Surface for NO2: Monthly Scaling of a Satellite-Derived Land-Use Regression, 2000–2010, Environ. Sci. Technol., 49, 12 297–12 305, doi:10.1021/acs.est.5b02882, URL https://doi.org/10.1021/acs.est.5b02882, 2015.

Bucsela, E., Celarier, E., Wenig, M., Gleason, J., Veefkind, J., Boersma, K., and Brinksma, E.: Algorithm for $NO_2$ vertical column retrieval from the ozone monitoring instrument, IEEE Trans. Geosci. Remote Sens., 44, 1245–1258, doi:10.1109/tgrs.2005.863715, URL https://doi.org/10.1109/tgrs.2005.863715, 2006.

Bucsela, E., Krotkov, N., Celarier, E., Lamsal, L., Swartz, W., Bhartia, P., Boersma, K., Veefkind, J., Gleason, J., and Pickering, K.: A new tropospheric and stratospheric $NO_2$ retrieval algorithm for nadir-viewing satellite instruments: applications to OMI, Atmos. Meas. Tech., 6, 2607–2626, doi:10.5194/amt-6-2607-2013, 2013.

Burrows, J. P. and Chance, K. V.: SCIAMACHY and GOME: the scientific objectives, in: Proc. SPIE, vol. 1715, doi:10.1117/12.140201, 1993.

Burrows, J. P., Richter, A., Dehn, A., Deters, B., Himmelmann, S., Voigt, S., and Orphal, J.: Atmospheric remote-sensing reference data from GOME–2. Temperature-dependent absorption cross section of $O_3$ in the 231-794 nm range, J. Quant. Spectrosc. Radiat. Transfer, 61, 509–517, 1999a.

Burrows, J. P., Weber, M., Buchwitz, M., Rozanov, V., Ladstätter-Weißenmayer, A., Richter, A., DeBeek, R., Hoogen, R., Bramstedt, K., Eichmann, K.-U., and Eisinger, M.: The Global Ozone Monitoring Experiment (GOME): Mission Concept and First Scientific Results, J. Atmos. Sci., 56, 151–175, doi:10.1175/1520-0469(1999)056⟨0151: TGOMEG⟩2.0.CO;2, 1999b.

Canty, T. P., Hembeck, L., Vinciguerra, T. P., Anderson, D. C., Goldberg, D. L., Carpenter, S. F., Allen, D. J., Loughner, C. P., Salawitch, R. J., and Dickerson, R. R.: Ozone and $NO_x$ chemistry in the eastern US: evaluation of CMAQ/CB05 with satellite (OMI) data, Atmos. Chem. Phys., 15, 10 965–10 982, doi:10.5194/acp-15-10965-2015, URL https://doi.org/10.5194/acp-15-10965-2015, 2015.

Chauhan, A., Krishna, M., Frew, A., and Holgate, S.: Exposure to nitrogen dioxide ($NO_2$) and respiratory disease risk, Rev. Environ. Health, 13, 73–90, URL https://www.scopus.com/inward/record.uri?eid=2-s2.0-0031842694&partnerID=40&md5=129866ddd76a17bb4f29b71e09ca636e, cited By 60, 1998.

Choi, S., Joiner, J., Choi, Y., Duncan, B. N., Vasilkov, A., Krotkov, N., and Bucsela, E.: First estimates of global free-tropospheric $NO_2$ abundances derived using a cloud-slicing technique applied to satellite observations from the Aura Ozone Monitoring Instrument (OMI), Atmos. Chem. Phys., 14, 10 565–10 588, doi:10.5194/acp-14-10565-2014, URL http://www.atmos-chem-phys.net/14/10565/2014/, 2014.

de Foy, B., Lu, Z., Streets, D. G., Lamsal, L. N., and Duncan, B. N.: Estimates of power plant NOx emissions and lifetimes from OMI NO2 satellite retrievals, Atmos. Environ., 116, 1–11, doi:10.1016/j.atmosenv.2015.05.056, URL https://doi.org/10.1016/j.atmosenv.2015.05.056, 2015.

Gorshelev, V., Serdyuchenko, A., Weber, M., Chehade, W., and Burrows, J. P.: High spectral resolution ozone absorption cross-sections—Part 1: Measurements, data analysis and comparison with previous measurements around 293 K, Atmos. Meas. Tech., 7, 609–624, doi:10.5194/amt-7-609-2014, URL https://www.atmos-meas-tech.net/7/609/2014/, 2014.

Hudman, R. C., Jacob, D. J., Turquety, S., Leibensperger, E. M., Murray, L. T., Wu, S., Gilliland, A. B., Avery, M., Bertram, T. H., Brune, W., Cohen, R. C., Dibb, J. E., Flocke, F. M., Fried, A., Holloway, J., Neuman, J. A., Orville, R., Perring, A., Ren, X., Sachse, G. W., Singh, H. B., Swanson, A., and Wooldridge, P. J.: Surface and lightning sources of nitrogen oxides over the United States: Magnitudes, chemical evolution, and outflow, J. Geophys. Res. Atmos., 112, doi:10.1029/2006JD007912, 2007.

Hudman, R. C., Moore, N. E., Mebust, A. K., Martin, R. V., Russell, A. R., Valin, L. C., and Cohen, R. C.: Steps towards a mechanistic model of global soil nitric oxide emissions: implementation and space based-constraints, Atmos. Chem. Phys., 12, 7779–7795, doi:10.5194/acp-12-7779-2012, URL https://www.atmos-chem-phys.net/12/7779/2012/, 2012.

Jiang, Z., McDonald, B. C., Worden, H., Worden, J. R., Miyazaki, K., Qu, Z., Henze, D. K., Jones, D. B. A., Arellano, A. F., Fischer, E. V., Zhu, L., and Boersma, K. F.: Unexpected slowdown of US pollutant emission reduction in the past decade, PNAS, 115, 5099–5104, doi:10.1073/pnas.1801191115, URL https://doi.org/10.1073/pnas.1801191115, 2018.

Kagawa, J.: Evaluation of biological significance of nitrogen oxides exposure, Tokai J. Exp. Clin. Med., 10, 348, 1985.

Kampa, M. and Castanas, E.: Human health effects of air pollution, Environ. Pollut., 151, 362 – 367, doi:https://doi.org/10.1016/j.envpol.2007.06.012, URL http://www.sciencedirect.com/science/article/pii/S0269749107002849, proceedings of the 4th International Workshop on Biomonitoring of Atmospheric Pollution (With Emphasis on Trace Elements), 2008.

Kharol, S., Martin, R., Philip, S., Boys, B., Lamsal, L., Jerrett, M., Brauer, M., Crouse, D., McLinden, C., and Burnett, R.: Assessment of the magnitude and recent trends in satellite-derived ground-level nitrogen dioxide over North America, Atmos. Environ., 118, 236–245, doi:10.1016/j.atmosenv.2015.08.011, URL https://doi.org/10.1016/j.atmosenv.2015.08.011, 2015.

Kiehl, J. and Solomon, S.: On the Radiative Balance of the Stratosphere, J. Atmos. Sci., 43, 1525–1534, doi:10.1175/1520-0469(1986)043⟨1525:OTRBOT⟩2.0.CO;2, 1986.

Laughner, J., Zhu, Q., and Cohen, R.: Berkeley High Resolution (BEHR) OMI NO2 - Gridded pixels, daily profiles, v3, UC Berkeley Dash, Dataset, doi:10.6078/D12D5X, 2018a.

Laughner, J., Zhu, Q., and Cohen, R.: Berkeley High Resolution (BEHR) OMI NO2 - Native pixels, daily profiles, UC Berkeley Dash, Dataset, doi:10.6078/D1WH41, 2018b.

Laughner, J., Zhu, Q., and Cohen, R.: Berkeley High Resolution (BEHR) OMI NO2 - Gridded pixels, monthly profiles, UC Berkeley Dash, Dataset, doi:10.6078/D1RQ3G, 2018c.

Laughner, J., Zhu, Q., and Cohen, R.: Berkeley High Resolution (BEHR) OMI NO2 - Native pixels, monthly profiles, UC Berkeley Dash, Dataset, doi:10.6078/D1N086, 2018d.

Laughner, J. L. and Cohen, R. C.: Quantification of the effect of modeled lightning $NO_2$ on UV-visible air mass factors, Atmos. Meas. Tech. Discuss., 2017.

Laughner, J. L., Zare, A., and Cohen, R. C.: Effects of daily meteorology on the interpretation of space-based remote sensing of $NO_2$, Atmos. Chem. Phys., 16, 15 247–15 264, doi: 10.5194/acp-16-15247-2016, URL http://www.atmos-chem-phys.net/16/15247/2016/, 2016.

Laughner, J. L., Zhu, Q., and Cohen, R.: Evaluation of version 3.0B of the BEHR OMI $NO_2$ product, Atmos. Meas. Tech. Discuss., 2018, 1–25, doi:10.5194/amt-2018-248, URL https://www.atmos-meas-tech-discuss.net/amt-2018-248/, 2018e.

Levelt, P., van der Oord, G., Dobber, M., Mälkki, A., Visser, H., de Vries, J., Stammes, P., Lundell, J., and Saari, H.: The Ozone Monitoring Instrument, IEEE Trans. Geosci. Remote Sense., 44, 1093–1101, doi:10.1109/TGRS.2006.872333, 2006.

McLinden, C. A., Fioletov, V., Boersma, K. F., Kharol, S. K., Krotkov, N., Lamsal, L., Makar, P. A., Martin, R. V., Veefkind, J. P., and Yang, K.: Improved satellite retrievals of $NO_2$ and $SO_2$ over the Canadian oil sands and comparisons with surface measurements, Atmos. Chem. Phys., 14, 3637–3656, doi:10.5194/acp-14-3637-2014, 2014.

Noel, S., Bovensmann, H., Burrows, J. P., Frerick, J., Chance, K. V., Goede, A. P. H., and Muller, C.: SCIAMACHY instrument on ENVISAT-1, in: Sensors, Systems, and Next-Generation Satellites II, edited by Fujisada, H., SPIE, doi:10.1117/12.333621, URL https://doi.org/10.1117/12.333621, 1998.

Parker, L., Kemball-Cook, S., and Yarwood, G.: Final Report Hood County $NO_x$ Trends, Tech. rep., URL http://www.hoodcountycleanair.com/userfiles/file/HoodCountyNOxTrends_06Oct2017.pdf, contract #582-16-60185, 2017.

Pusede, S. E., Duffey, K. C., Shusterman, A. A., Saleh, A., Laughner, J. L., Wooldridge, P. J., Zhang, Q., Parworth, C. L., Kim, H., Capps, S. L., Valin, L. C., Cappa, C. D., Fried, A., Walega, J., Nowak, J. B., Weinheimer, A. J., Hoff, R. M., Berkoff, T. A., Beyersdorf, A. J., Olson, J., Crawford, J. H., and Cohen, R. C.: On the effectiveness of nitrogen oxide reductions as a control over ammonium nitrate aerosol, Atmos. Chem. Phys., 16, 2575–2596, doi:10.5194/acp-16-2575-2016, URL https://doi.org/10.5194/acp-16-2575-2016, 2016.

Russell, A., Perring, A., Valin, L., Bucsela, E., Browne, E., Min, K., Wooldridge, P., and Cohen, R.: "A high spatial resolution retrieval of $NO_2$ column densities from OMI: method and evalutation", Atmos. Chem. Phys., 11, 8543–8554, doi:10.5194/acp-11-8543-2011, 2011.

Russell, A. R., Valin, L. C., and Cohen, R. C.: Trends in OMI $NO_2$ observations over the United States: effects of emission control technology and the economic recession, Atmos. Chem. Phys., 12, 12 197–12 209, doi:10.5194/acp-12-12197-2012, 2012.

Travis, K. R., Jacob, D. J., Fisher, J. A., Kim, P. S., Marais, E. A., Zhu, L., Yu, K., Miller, C. C., Yantosca, R. M., Sulprizio, M. P., Thompson, A. M., Wennberg, P. O., Crounse, J. D., St. Clair, J. M., Cohen, R. C., Laughner, J. L., Dibb, J. E., Hall, S. R., Ullmann, K., Wolfe, G. M., Pollack, I. B., Peischl, J., Neuman, J. A., and Zhou, X.: Why do models overestimate surface ozone in the Southeast United States?, Atmos. Chem. Phys., 16, 13 561–13 577, doi:10.5194/acp-16-13561-2016, URL https://www.atmos-chem-phys.net/16/13561/2016/, 2016.

Veefkind, J., Aben, I., McMullan, K., Förster, H., de Vries, J., Otter, G., Claas, J., Eskes, H., de Haan, J., Kleipool, Q., van Weele, M., Hasekamp, O., Hoogeveen, R., Landgraf, J., Snel, R., Tol, P., Ingmann, P., Voors, R., Kruizinga, B., Vink, R., Visser, H., and Levelt, P.: TROPOMI on the ESA Sentinel-5 Precursor: A GMES mission for global observations of the atmospheric composition for climate, air quality and ozone layer applications, Remote Sens. Environ., 120, 70–83, doi:10.1016/j.rse.2011.09.027, 2012.

Wegmann, M., Fehrenbach, A., Heimann, S., Fehrenbach, H., Renz, H., Garn, H., and Herz, U.: $NO_2$-induced airway inflammation is associated with progressive airflow limitation and development of emphysema-like lesions in C57BL/6 mice, Exp. Toxicol. Pathol, 56, 341–350, doi:10.1016/j.etp.2004.12.004, URL https://www.scopus.com/inward/record.uri?eid=2-s2.0-17844400917&doi=10.1016%2fj.etp.2004.12.004&partnerID=40&md5=25fa3a07d2fd636156eba84096769762, 2005.

---

## Author Comment (AC2) · 12 Sep 2018

**The Berkeley High Resolution Tropospheric NO$_2$ Product**

**Response to Anonymous Referee #2**

Joshua L. Laughner, Qindan Zhu, and Ronald C. Cohen

September 11, 2018

We thank the reviewer for their careful reading and especially their critical thinking about the explanation of the difference between average VCDs using daily and monthly profiles and our explanation of the root cause for the difference between the new and old visible AMF formulation. We have reexamined our conclusions in both cases and made edits where appropriate.

Responses to specific comments follow. The reviewer's comments will be shown in red, our response in blue, and changes made to the paper are shown in black block quotes. Unless otherwise indicated, page and line numbers correspond to the original paper. Figures, tables, or equations referenced as "R$n$" are numbered within this response; if these are used in the changes to the paper, they will be replaced with the proper number in the final paper. Figures, tables, and equations numbered normally refer to the numbers in the original discussion paper.

However, I have got the feeling, at least from the abstract, that the manuscript is more suitable to journals like AMT or ACP, since it is mainly talking about algorithm instead of the dataset itself. Therefore, I would suggest the authors to update the abstract and maybe also the conclusion with more descriptions about the dataset...

Because the BEHR product will continue to evolve as we learn more about the ideal design of high-resolution NO$_2$ retrievals, we believe that ESSD's living data process is an ideal way to communicate those updates. For a satellite NO$_2$ product, where the assumptions and a priori data about the atmosphere can have a significant impact on the final NO$_2$ VCDs, we feel that it is important to show how the evolution of these assumptions impacts the VCDs. Nevertheless, this is a fair comment, as we also have a responsibility to describe the dataset for the users. To this end, we have expanded both the section on usage recommendations, to give users some guidance on typical uses for BEHR data, and added more information about the primary variables in the Appendix:

"

**General recommendations**

**Quality filtering**

It is vital in any use of BEHR data to filter out low quality data. The BEHR algorithm attempts to calculate an $NO_2$ VCD for as many pixels as possible, even if some of those pixels are known to be poor quality. The philosophy is that it is better to have data for a pixel if at all possible and remove it only if the quality is too low for a particular application. Some causes of low quality (e.g. the row anomaly, `https://projects.knmi.nl/omi/research/product/rowanomaly-background.php`) make the $NO_2$ column unusable under any case, while others (e.g. high cloud fraction, low quality surface reflectance) only affect certain uses.

The quality of the pixel is summarized in the first two (least-significant) bits of the BEHRQualityFlags field. The second bit is a critical error bit, if set (i.e. if a bitwise AND of BEHRQualityFlags with 2 is $> 0$) then the $NO_2$ columns for that pixel should not be used under any conditions. The first bit is a quality flag bit; if it is set (if a bitwise AND of BEHRQualityFlags with 1 is $> 0$) then the use of the column for typical applications wanting information down to the surface is not recommended; however, other applications may still find use for this pixel. For example, the first bit is set if the OMI geometric cloud fraction is $> 0.2$, since the uncertainty of the total tropospheric column increases greatly as more $NO_2$ is obscured by clouds, but cloud slicing approaches (e.g. Choi et al., 2014; Marais et al., 2018) will actually prefer large cloud fractions, and so will need to do their own cloud filtering. For most applications however, it is recommended to ignore pixels that have the first (i.e. quality summary) bit set to 1.

Users must also be sure to remove fill values. The fill value for each field is defined in the "fillvalue" attribute. Generally, checking if a value is exactly equal to a fill value is not recommended unless the value is an integer type, as floating point error on some systems may cause fill values to be missed. It is better practice to check for values within some relative tolerance of the fill value:

$$|x - f| < |f| \cdot t \tag{R1}$$

where $x$ is the data, $f$ is the fill value, and $t$ the tolerance. $t = 10^{-4}$ works in our experience.

**Choice of daily or monthly profile subproduct**

Users will also need to choose whether to use the subproduct with daily profiles. Use of the subproduct with daily profiles is strongly encouraged if possible, for two reasons. First, the daily profiles also use year specific emissions (Sect. 2.6.1, so will better capture trends in VCDs as the surface contribution to the a priori profiles is reduced. Second, Laughner et al. (2016) showed that using daily profiles

significantly changes day-to-day VCDs, and that some applications of satellite data can be biased when monthly profiles are used. Applications similar to those studied in Laughner et al. (2016), where upwind or downwind columns are systematically averaged together are particularly vulnerable to bias when monthly average profiles are used.

Caution is advised if comparing 2005 or 2006 data using daily profiles to other years; the different WRF-Chem boundary conditions (Sect. 2.6.1) may also bias observed trends. This effect is likely small, as in a test of 1 week of data using two sets of profiles, one using GEOS-Chem boundary conditions and one using MOZART boundary conditions, the mean change was $< 10^{14}$ molec cm$^{-2}$, and only 0.7% of pixels with any cloud fraction had a change exceeding $10^{15}$ molec cm$^{-2}$ (0.05% of pixels with cloud fraction $< 0.2$).

However, mixing daily and monthly profile subproducts is *strongly* discouraged, as systematic differences between them (i.e. Sect. 4.3.2 of this paper; Laughner et al., 2016) will bias any trends observed.

**Application #1: direct observation of VCDs**

Direct observation of VCDs has a number of applications, including elucidating trends in NO$_2$ burdens (e.g. Russell et al., 2012; Jiang et al., 2018) or inferring lightning emissions (e.g. Pickering et al., 2016). Users wanting to average BEHR data over a given time period, e.g. to compare summer average NO$_2$ columns for different years, will find this easiest using the gridded data, as this places the NO$_2$ columns on a consistent equirectangular latitude/longitude grid (i.e. the data in grid cell (1,1) will be at the same lat/lon in each orbit, whereas in the native data, pixel (1,1) will not), so it is easy to average across different days. When averaging, each grid cell should be weighted by the Areaweight value given in the gridded product; this is the inverse of the pixel area, so weighting by this inherently gives more weight to smaller, more representative pixels.

Users interested in VCDs from individual days (e.g. to find NO$_2$ downwind of an episodic event such as lightning) can use either the native pixel or gridded products, whichever is easier. In this case, it is important to keep in mind that pixel sizes vary from day-to-day. Therefore, if the source signal of interest is smaller than a single pixel, it will be more diluted if it falls in a larger pixel on the edge of the OMI swath than a small one near the center.

**Application #2: inferring surface NO$_2$ concentration**

Since a VCD is a measurement integrated over the troposphere, it does not directly provide information about the surface concentration of NO$_2$. The simplest approach to infer ground-level NO$_2$ concentrations from VCDs is to multiply the BEHR VCD by the ratio of surface concentration to VCD obtained from a modeled NO$_2$ profile (Lamsal et al., 2008):

$$[NO_2]_{surf} = \frac{g(p_{surf})}{\int_{p_{surf}}^{p_{trop}} g(p)\, dp} V_{BEHR} \qquad (R2)$$

where $g(p)$ is the modeled profile, $p_{surf}$ the surface press, $p_{trop}$ the tropopause pressure, and $V_{BEHR}$ the BEHR VCD. $g(p)$ may be obtained in many ways; for users without model output or measurements of $NO_2$ profiles, the a priori profiles used in BEHR are included in the native pixel subproduct and may be used for this purpose. In this case, using the subproduct with daily profiles is highly recommended so that the profiles respond to changes in meterology day to day, especially wind fields.

**Application #3: comparing to models**

Users wishing to compare BEHR VCDs to model output should follow the suggestions in Boersma et al. (2016). This requires calculating the overlap between the BEHR pixels and the user's model grid cells and applying the BEHR averaging kernel to the user's model profile before calculating the model VCD, so the native pixel product must be used, since it contains the averaging kernels and the pixel corners.

The averaging kernels would be applied to the model profile as:

$$V_{model} = \sum_{k} c_k a_k \qquad (R3)$$

where $V_{model}$ is the modeled VCD after applying the averaging kernels, $k$ is the level index, $c_k$ is the model profile converted to a partial column for level $k$, and $a_k$ is the averaging kernel for level $k$.

There are three important considerations in this application. First, since BEHR provides only a tropospheric VCD, it must be compared against a modeled tropospheric column, no stratospheric component may be included.

Second, the model $NO_2$ profile should be interpolated to the pressure levels on which the averaging kernels are defined (given in the BEHR files as BEHRPressureLevels) rather than the other way around. This is because the averaging kernels may have sharp changes between levels (usually at the cloud pressure, since OMI's sensitivity increases dramatically over a bright cloud) so interpolating the averaging kernels to the model pressures is more likely to introduce errors.

Third, the model profile is best converted to partial columns before applying the averaging kernels. This may be done several ways, such as:

- Interpolate the profile to the averaging kernels' pressure levels, then multiply the profile concentration as number density by the layer height.

- Interpolate the profile to the *edges* of the averaging kernels' levels, then integrate over each layer to obtain the partial column.

Both methods need the edge of the pressure levels, either to calculate the box height or to define the limits of the integration. Since the pressures given for the averaging kernels are the level centers, the edges are most easily defined as the midpoints between those layers; with the surface pressure serving as the lower limit of the bottom layer and the tropopause pressure serving as the upper limit of the top layer.

Converting from pressure to altitude for either method can either be done using a scale height relation (e.g. Eq. 10), though this will likely introduce some error as we saw in Sect. 5.4 that the meteorological correction can be significant. A better option, if the user's model output includes altitude and pressure vectors, is to interpolate the altitude from the model to the averaging kernels' pressure levels alongside the $NO_2$. Alternatively, in the second method, $NO_2$ profiles in mixing ratio can be directly integrated over pressure (Ziemke et al., 2001, Appendix B). This is done internally in BEHR using the integPr2 code at `https://github.com/CohenBerkeleyLab/BEHR-core-utils/blob/develop/AMF_tools/integPr2.m`.

**(Appendix) Key variables**

The BEHR files contain a large number of variables, including a large amount of ancillary data used in the algorithm. All variables in the HDF files have a "description" attribute that provides some information about what they are. They also have a "product" attribute that indicates whether they are taken verbatim from the NASA Standard Product (product = "SP") or added by BEHR (product = "BEHR"). The primary variables that most users should focus on are:

- **BEHRColumnAmountNO2Trop**: This is the tropospheric VCD calculated using Eqs. (1) and (2). It is the concentration of $NO_2$ integrated from the surface to the tropopause, including $NO_2$ below clouds. This is the $NO_2$ value that most users should use.

- **BEHRColumnAmountNO2TropVisOnly**: This is the visible-only tropospheric VCD calculated with Eqs. (1) and (3). It excludes below-cloud $NO_2$. Generally the use for this quantity is more specialized; most users should use the previous value.

- **BEHRQualityFlags**: A 32-bit unsigned integer value where each bit represents a boolean flag indicating the presence of a specific error or warning for that pixel. See Sect. A3 for details.

- **Areaweight** (*gridded products only*): a weight calculated of the inverse of the area of the pixel that each grid cell falls within. This should be used to weight the gridded data during temporal averaging (see Sect. 7).

- **Longitude**, **Latitude**: the coordinates of the pixel or grid cell center.

- **CloudFraction**: this is a geometric cloud fraction from the OMI $O_2-O_2$ cloud product (Acarreta et al., 2004). It is the default used to filter for cloudy pixels, and is the same as the corresponding variable in the NASA Standard Product.

- **CloudRadianceFraction**: this is a radiance cloud fraction (i.e. one weighted by the amount of light coming from the cloud vs. the ground). It is the same as the corresponding field in the NASA Standard Product.

- **MODISCloud**: this is a geometric cloud fraction from the Aqua MODIS instrument (Platnick et al., 2015) averaged to the OMI pixels. It is an alternate way of filtering for cloudy pixels that may be less susceptible to false positives from highly reflective ground (Russell et al., 2011). Some pixels near the edge of the swath may be missing this data since the MODIS swath width is slightly smaller than OMI's.

More advanced users may find the 3D variables included in the native pixel sub-products useful. These variables give a unique vector of values for each pixel. In Matlab, the vector for each pixel runs along the first dimension, so if the $NO_2$ VCDs are the 2D array `V` and one of the 3D arrays is `A`, then the vector corresponding to `V(i,j)` would be `A(:,i,j)`. However, some languages reverse the order of the dimensions. In BEHR v3.0B, the vector dimension can be identified as the one with a length of 33.

In BEHR, these 3D variables are defined on a vertical grid of 30 standard pressure levels (ranging from 1020 to 60 hPa) with values interpolated to the surface pressure, cloud pressure, and tropopause pressure included, brining the total length of the vertical dimension to 33. If one of the interpolated pressure levels is the same as a standard pressure level, the value is not duplicated, and the vector of values will be padded with fill values at the end.

- **BEHRPressureLevels**: this dataset defines the pressure levels that the other 3D variables are defined on.

- **BEHRNO2apriori**: this dataset gives the $NO_2$ a priori profiles used in the BEHR retrieval in mixing ratio.

- **BEHRAvgKernels**: these are the averaging kernels referenced in Sect. 7.4. They are defined as:

$$a(p) = \frac{(1-f)w_{\text{clear}}(p)\alpha(p) + f w_{\text{cloudy}}(p)\alpha(p)}{A} \tag{R4}$$

where $a(p)$ is the averaging kernel, $f$ the cloud radiance fraction, $\alpha(p)$ the temperature correction (Eq. 6) and $w_{\text{clear}}(p)$ and $w_{\text{cloudy}}(p)$ the clear and cloudy scattering weights, which are set to 0 below the surface and cloud pressure, respectively.

- **BEHRScatteringWeightsClear**, **BEHRScatteringWeightsCloudy**: the temperature corrected clear and cloudy scattering weights, set to 0 below the surface and cloud pressure, respectively, i.e.:

$$w'_{\text{clear}}(p) = w_{\text{clear}}(p)\alpha(p) \qquad \qquad \text{(R5)}$$

$$w'_{\text{cloudy}}(p) = w_{\text{cloudy}}(p)\alpha(p) \qquad \qquad \text{(R6)}$$

"

general: Despite the good written language, the organization sometimes makes it difficult for me to identify which improvement is for v3.0A which is for v3.0B. For instance, in the "Methods" section, some methods are introduced for v3.0A and some are for v3.0B. Also, the methods of older version are sometimes introduced in "Methods" section (e.g. surface pressure) and sometimes in each subsection (e.g. visible-only AMF calculation). Therefore, this methods section is not fully referable when reading the following sections. In addition, illustrations like "figure 1 shows..." and "table 1 shows..." are missing.

We have added references to figures and tables in the relevant sections. Most notably, the main figures (1 and 4 in the discussion paper) are introduced in a new "Paper structure" section that comes before the detailed analysis of the results.

"In sections 4 and 5, we evaluate the effect each change to the BEHR algorithm between v2.1C and v3.0B had on the tropospheric VCDs. In order to provide a clear history, changes introduced in v3.0A will be discussed first (Sect. 4), followed by changes introduced in v3.0B (Sect. 5). V3.0A incorporated all changes up through the introduction of the new gridding algorithm; the remainder are added in v3.0B. Changes to the visible-only VCDs (i.e. those excluding the below-cloud column) are discussed in the supplement (Sect. S1). Following this the overall difference between v2.1C and v3.0B will be presented in Sect. 6. Recommendations for the use of the product are given in Sect. 7. A description of the data format is given in Appendix A.

For the discussion of how changes to the algorithm affect the $NO_2$ VCDs, figures 1 and 2 and Tables 3 and 4 are the central focus. Each panel shows the change in the BEHR $NO_2$ VCDs resulting from a specific change to the algorithm. To generate these figures, BEHR VCDs were computed after adding each change to the algorithm incrementally. Each panel in the figures and line in the tables shows the percent change in VCDs due to the corresponding change to the algorithm. These are computed relative to VCDs with one fewer change to the algorithm; for example, Fig. 1b is the percent difference between VCDs using the new NASA SCDs and the new MODIS BRF surface reflectance versus VCDs using just the new NASA SCDs. The (a) panels in Figs. 1 and 2 and the first lines in Tables 3 and 4 are relative to BEHR v2.1C.

Figure 1 shows the percent change of average BEHR tropospheric VCDs due to each algorithm improvement for the subproduct using monthly average $NO_2$ a priori profiles, while Fig. 2 shows the changes to the subproduct using daily $NO_2$ a priori profiles. (Figure 2 has fewer panels than Fig. 1 as daily profiles were only possible in increments after the change to the algorithm to introduce the new a

| Component | v3.0A | v3.0B | Section |
|---|---|---|---|
| Ocean reflectance | Calc. for 430 nm | Calc. for 460 nm | 2.2 |
| Surface pressure | Scale height | WRF pressure adjusted with GLOBE elevation | 2.3 |
| Tropopause pressure | Fixed at 200 hPa | Calculated from WRF temperature profiles | 2.4 |
| Daily prof. hour | Last hour before overpass | Closest hour to overpass | 2.6.2 |

Table R1: Summary of differences in methods between v3.0A and v3.0B.

priori profiles was implemented.) Both figures are for summer (June–Aug.) 2012. Winter changes are presented in the supplement.

Table 3 gives the mean and median changes for each incremental improvement shown in Figs. 1 and 2; that is, it gives the domain-wide mean and median values of the time-averaged changes shown in the figures. Table 4 is similar, but is the statistics for individual pixels, rather than the time-averaged changes."

For the methods section, we have generally edited it to make clear which methods are applicable to v3.0A and which to v3.0B, including a summary table:

The visible-only AMF given in the methods is the only form used in both v3.0A and v3.0B, so including the old form in the methods for v3.0x might add confusion. We have added a sentence referring the reader to the appropriate section:

"Replacing $A_{\mathrm{BEHR}}$ in Eq. (1) with $A_{\mathrm{BEHR,vis}}$ yields a visible-only $NO_2$ column as the output, stored in the variable "BEHRColumnAmountNO2TropVisOnly" in the BEHR files. **The form of this visible AMF changed from v2.1C to v3.0A; please see Sect. S1 in the Supplement for details of the old calculation.**"

""

page2line4 GOME2 (GOME2A in 2006 and GOME2B in 2012) is newer than OMI. We have removed GOME2 from this sentence:

"The spatial resolution available with early instruments (i.e. the Global Ozone Monitoring Experiment, GOME, $40 \times 320$ km$^2$, Burrows et al. 1999b; the SCanning Imaging Absorption SpectroMeter for Atmospheric CHartographY, SCIA-MACHY, $30 \times 60$ km$^2$, Noel et al. 1998) allowed inferences at the scale of entire continents or entire metropolitan regions..."

page2line24 I believe the difficulty of NO observation is not only because of the absorption in UV. Even it is noisier than VIS, it still works for gases, e.g. O3 and even possible for NO2. Please specify this sentence.

We believe that the difficulty in measuring tropospheric NO specifically is that the absorbance features overlap with the strong ozone absorption band, so NO absorption in the troposphere is obscured. However, we have not found any citations that explicitly state this, so for simplicity we have altered this to just note that NO is not measured by any sensors currently in orbit:

"The current fleet of space-based sensors measures $NO_2$, not total $NO_x$, but due to the rapid daytime equilibrium between NO and $NO_2$, this allows inferences about tropospheric $NO_x$ to be made from $NO_2$ measurements."

page2line25 "inferences about total NOx are made from NO2 measurements" maybe also because of the quick conversion of NO to NO2?
We have added:

"In contrast, $NO_2$ has useful absorbance in the visible wavelengths, outside the strong ozone absorption band. In combination with the rapid daytime equilibrium between NO and $NO_2$, this allows inferences about tropospheric $NO_x$ to be made from $NO_2$ measurements."

page3line4 I suggest including the TM5 also as examples, since it is largely used currently, e.g. for OMI and TROPOMI retrieval.
We have added TM5 to this list.

page3line3-11 I recommend combining these two paragraphs together, since they both talk about how to calculate AMFs, and the "input data" in line 12 talks mainly about the input data (i.e. profiles) in the 1st paragraph.
We have combined these two paragraphs as suggested.

page5line8 The definition of RRA it not that special, can be removed.
We have found that different communities use different conventions for RAA; some define 0 as the sun and viewer are on the same side, others define it as the sun and viewer are directly opposite each other. We prefer to be explicit about how we define RAA so there is no confusion, and this seems the logical place to do so.

page6line10 What does "BEHR uses the file dated for the day being retrieved for the BRF coefficients." mean?
We have clarified this with an example:

"BEHR uses the file dated for the day being retrieved for the BRF coefficients, **i.e. for 1 June 2012, the MODIS files with 1 June 2012 in the file name are used. This means that the surface reflectivity used in BEHR incorporates land data from 8 days before and after the OMI observation.**"

page8line7 Why does the BEHR include these different cloud products? For instance, is there a specific reason to include MODIS cloud fraction? Please also add more information about the OMI-derived quantities. Do you retrieve the cloud fractions, or do you use OMCLDO2 or OMCLDRR?
We have added:

"BEHR contains several cloud fraction products: **a geometric cloud fraction derived from the O$_2$-O$_2$ algorithm (Acarreta et al., 2004), a cloud radiance fraction calculated by NASA from the O$_2$-O$_2$ product,** and a geometric cloud fraction derived from the Aqua MODIS instrument, and cloud pressure from the OMI O$_2$-O$_2$ algorithm (Acarreta et al., 2004). The OMI-derived quantities are the same as those in the NASA SP v3.0. The MODIS cloud product used is MYD06_L2 (Platnick et al., 2015).

**Russell et al. (2011) found that the MODIS cloud product was less likely to give erroneously large cloud fractions due to high surface reflectivity over the California and Nevada desert, and concluded that this more than offset any error caused by the small separation between the overpass times (currently $\sim 8$ min) of OMI on board the Aura satellite and MODIS on board the Aqua satellite. We continue to provide the MODIS cloud product for cloud filtering; however, because it does not cover the full OMI swath, we use the OMI cloud fractions in the AMF calculations.**"

Also, please update the expression "radiance cloud fraction" to cloud radiance fraction here and through the manuscript.
So updated.

page8line28 What output is used here?
The chemical concentrations are used for WRF-Chem boundary conditions. We have clarified as:

"**Chemical boundary conditions for WRF-Chem are taken two different global models.** For model years 2007 and later, **chemical concentrations** from the Model for Ozone and Related chemical Tracers (MOZART, Emmons et al., 2010) provided by the National Center for Atmospheric Research"

page9line7 When would this extrapolation happen? If it is because of the different surface pressure from scattering weight and profiles, then it might be even better to shift the profile but not extrapolate.
Yes, this happens if the surface pressure of the pixel is below the bottom of the profile. Shifting the profile is an interesting idea, though if large shifts were required, it could possibly introduce different errors by e.g. placing the boundary layer or upper tropospheric lightning at the wrong altitudes. One thing we had not mentioned originally is that the extrapolation is only allowed to extend the profile by one pressure level. Near the surface the pressure levels are quite close together (5 hPa) and even at 1 km elevation the spacing is only 25 hPa, so we expect the error to be minimal. We have added text describing this:

"**The profiles are also extrapolated to one scattering weight pressure level above and below the top and bottom of the WRF profile, respectively. This accounts for the possibility that e.g. a pixel's surface**

**pressure may be slightly below the WRF surface pressure, but by limiting the extrapolation to only one level, should minimize errors due to extrapolation.** Once interpolated and extrapolated, all profiles within the..."

page9line14 Please specify "when possible".
Changed to:

"We make use of daily profiles **for as much of the OMI data record as it is computationally feasible to simulate these profiles.** "

page11table2 What is ocean LUT here and through the manuscript? Do you mean ocean reflectance LUT?
Yes, we have clarified this in the caption and elsewhere.

page11figure2 This figure is comparing with BEHR v3.0B but it is described in the "Changes in BEHR v3.0A" text. Additionally, the interpretation after figure is talking about the changes in surface reflectance over land. Please add more analysis about changes over water, since the difference is quite significant.
We have made the comparison with v3.0A instead. We also incorporated the decomposition of changes due to the change of MODIS version vs. black-sky to BRF in response to a comment from the first reviewer, as well as added a panel showing the ocean LUT and added the following text:

"

BEHR v2.1C used an ocean reflectance look-up table embedded in the core code that defined how the dependence of the ocean reflectance on solar zenith angle (SZA). As documentation of the source for this table is not available, BEHR v3.0A switched to a new look up table calculated explicitly using the Coupled Ocean-Atmosphere Radiative Transfer (COART) model (Jin et al., 2006). The difference in the SZA dependence of the look up tables is shown in Fig. 3g. The overall shape is similar, but the difference between small and large SZAs is less pronounced in the new ocean look-up table. Both are similar to the ocean surface reflectance calculated by Jin et al. (2004) for an atmospheric aerosol optical depth of 1, but for different wind speeds: the BEHR v2.1C look-up table is more characteristic of slow ($< 1$ m s$^{-1}$) winds, while the v3.0A table assumes a wind speed of 5 m s$^{-1}$.

At small SZAs characteristic of summer OMI observations ($< 35°$), the new look up table yields a $\sim 50\%$ greater ocean reflectance than the old table, which leads to the off-shore reflectance changes seen in Fig. 3a. At larger SZAs more characteristic of winter ($\sim 40°$ to $60°$), the difference between the old and new look-up tables shrinks, resulting in less change in the wintertime ocean surface reflectance (Fig. 3d).

Especially in summer, since the relative change in the ocean surface reflectance is large, using the new ocean look-up table does result in large relative changes

[Figure]

Figure R1: (a,d) Difference in surface reflectance between BEHR v2.1C (MODIS MCD43C3 black sky albedo, old ocean look up table) and BEHR v3.0B (MODIS MCD43Dxx BRF, new look up table) (b,e) Difference in surface reflectance between version 5 and 6 of the MODIS black sky albedo (no change in ocean look up table). (c,f) Difference in surface reflectance between the MODIS black sky and BRF product and the change in ocean look up table. (a–c) are for summer (JJA) and (d–f) are for winter (DJF). (g) The ocean albedo look-up table values for v2.1C, v3.0A, and v3.0B. (The change between v3.0A and v3.0B is discussed in Sect. 5.2.)

to the NO$_2$ VCDs. Along the coasts, these changes can reach $2 \times 10^{15}$ to $3 \times 10^{15}$ molec cm$^{-2}$ (or more near New York, NY), but away from the coasts, the absolute differences are quite small."

page13figure3 The figures are not clear and the conclusions are not convincing to me. Why is there a straight line (a deep blue line with no percent change in NO2 near cloud pressure 700 hPa in (a) and a distinct line near the black dashed line in (b))? What is the definition of surface NO2 concentration and why does it matter here? Since in Eq. 2, it is only the profile shape (relative vertical distribution) matters in the AMF cal- culation but not the absolute concentration. Also, I am not sure with "Greater percent difference with greater surface NO2 concentration.", because most of the largest differ- ences in the figure are found for low surface NO2 concentration (blue to deep blue in (a)). Similarly, greater percent differences are also found for small difference between the cloud fraction and cloud radiance fraction, since quite a lot of yellow dots are close to the black dashed line in (b).

We have reevaluated our conclusions here. We have revised this section and updated the figure. The line near the black dashed line in the original (b) panel was caused by pixels that had a high surface reflectivity in the NASA product ($\sim 0.55$), since when the ground is highly reflective, that reduces the discrepancy in light coming from the cloud vs. the ground, and so shrinks the difference between the geometric and radiance cloud fractions. Also please note that this section has been moved to the supplement, as in responding to a comment from Reviewer 1 we decided that the paper was more logically structured by including only the changes that impact the total tropospheric VCDs in the body of the main paper.

"As described in the methods (Sect. 2.1), BEHR has, since v2.1C, included both a total tropospheric NO$_2$ column and a "visible-only" column. Figure S1 provides a graphic definition of these terms. The visible-only column includes the NO$_2$ that would be visible if observing the pixel from directly above: for the cloudless part of the pixel, the column extends to the ground, but for the cloud covered part it only extends down to the cloud top. In contrast the standard total tropospheric column is the sum of the visible-only column and the ghost column, where the ghost column is the NO$_2$ below the clouds.

The AMF necessary to convert the observed slant columns to a visible-only ver- tical column (a what we will term a "visible-only" AMF) can be conceptualized two different ways. The formula for the v3.0 visible-only AMF is given in Eq. (3). Conceptually, this is the model SCD divided by the modeled VCD. In v2.1C, an alternate formulation was used:

$$A_{\mathrm{BEHR,vis}} = (1 - f)A_{\mathrm{clear,vis}} + f A_{\mathrm{cloudy,vis}} \qquad \text{(R7)}$$

where $f$ is again the cloud radiance fraction and

$$A_{\text{clear,vis}} = \frac{\int_{p_{\text{surf}}}^{p_{\text{trop}}} w_{\text{clear}}(p)g(p)\,dp}{\int_{p_{\text{surf}}}^{p_{\text{trop}}} g(p)\,dp} \tag{R8}$$

$$A_{\text{cloudy,vis}} = \frac{\int_{p_{\text{cloud}}}^{p_{\text{trop}}} w_{\text{cloudy}}(p)g(p)\,dp}{\int_{p_{\text{cloud}}}^{p_{\text{trop}}} g(p)\,dp} \tag{R9}$$

This earlier method assumes that each pixel can be treated as two totally independent subpixels, one clear and one cloudy. This seems a logical extension of the independent pixel approximation (Cahalan et al., 1994; Marshak et al., 1998), but the physical interpretation is less clear than the new formulation.

Although both approaches to calculating a visible-only AMF (i.e. Eq. R7 and Eq. 3) are conceptually valid , they are not mathematically identical, and so the retrieved visible tropospheric $NO_2$ column increases between v2.1C and v3.0A Figure S2 shows the average change in visible-only $NO_2$ columns when changing from the v2.1C AMF to the v3.0A AMF. In the summer (Fig. S2a) the average increase approaches 100% over the eastern US, decreasing to 0 towards the west coast. In the winter (Fig. S2b) the difference is more sporadic.

The main cause for the change is the difference in how the relative magnitude of the $NO_2$ to-ground VCD and the above-cloud VCD is treated by the AMF calculation. In the v2.1C visible-only AMF formulation, the relative contribution of the clear- and cloudy- sky AMFs was entirely determined by the cloud radiance fraction. Equation (R7) can be written as:

$$A_{\text{BEHR,vis}} = (1 - f)\frac{S_{\text{clear}}}{V_{\text{clear}}} + f\frac{S_{\text{cloudy}}}{V_{\text{cloudy}}} \tag{R10}$$

where $f$ is the cloud radiance fraction, $S_{\text{clear}}$ and $V_{\text{clear}}$ are the modeled slant and vertical $NO_2$ column density for the clear part of the pixel and $S_{\text{cloudy}}$ and $V_{\text{cloudy}}$ are likewise the modeled slant and vertical column density for the cloudy part of the pixel. $V_{\text{clear}}$ and $V_{\text{cloudy}}$ may be very different magnitudes (by a factor of up to 1000), especially in polluted areas where most of the $NO_2$ is near the surface and therefore below the cloud. However, the slant columns are related to their corresponding vertical columns through the scattering weights, which typically means the corresponding $S$ and $V$ values will be within about a factor of 2 or 3 of each other. This means that, in Eq. (R10), the relative magnitudes of $S_{\text{clear}}/V_{\text{clear}}$ versus $S_{\text{cloudy}}/V_{\text{cloudy}}$ will be similar, even if $V_{\text{clear}}$ and $V_{\text{cloudy}}$ (and likewise $S_{\text{clear}}$ and $S_{\text{cloudy}}$) are substantially different.

In contrast, the new formulation could be written as:

$$A_{\text{BEHR,vis,new}} = \frac{(1 - f)S_{\text{clear}} + fS_{\text{cloudy}}}{(1 - f_g)V_{\text{clear}} + f_gV_{\text{cloudy}}} \tag{R11}$$

[Figure]

Figure R2: **(a)** The percent change in the visible-only BEHR $NO_2$ VCD versus the ratio modeled VCDs ($V_{clear}$, the WRF-Chem profiles integrated over the whole tropo-sphere) to those integrated from cloud top to tropopause ($V_{cloudy}$), colored by cloud radiance fraction. **(b)** The ratio of $V_{clear}/V_{cloudy}$ versus geometric cloud fraction, cloud pressure, and colored by shape factor (mixing ratio/$V_{clear}$) at the surface. **(c)** The percent change in visible-only $NO_2$ VCD as a function of cloud radiance fraction and geometric cloud fraction. The black dashed line in (c) is the 1:1 line. *Note:* the color scale saturates at $10^{-24}$ in (b) and 100% in (c) to emphasize the distribution of the percent changes.

where $f_g$ is the geometric cloud fraction. In this case, the relative magnitudes of $S_{\text{clear}}$ versus $S_{\text{cloudy}}$ and $V_{\text{clear}}$ versus $V_{\text{cloudy}}$ does matter. If $V_{\text{cloudy}} \ll V_{\text{clear}}$, then Eq. R11 reduces to

$$A_{\text{BEHR,vis,new}} = \frac{(1-f)S_{\text{clear}}}{(1-f_g)V_{\text{clear}}} \tag{R12}$$

whereas in Eq. (R10), the second term does not go to zero when $V_{\text{cloudy}} \ll V_{\text{clear}}$ because $S_{\text{cloudy}} \propto V_{\text{cloudy}}$. This means that, in theory, when $V_{\text{cloudy}} \ll V_{\text{clear}}$, the new visible-only AMF will essentially be a clear sky AMF, which will be less than a cloudy sky AMF since it includes near-surface $NO_2$ that OMI is less sensitive to. In contrast, in the old formulation, the relative contribution of the clear and cloudy components only depends on the cloud radiance fraction, not the relative magnitude of $V_{\text{clear}}$ and $V_{\text{cloudy}}$, so the old visible-only AMFs will more often be of similar magnitude to a cloudy AMF. Because $V = S/A$ and $A_{\text{clear}} < A_{\text{cloudy}}$ in most cases, this means that the new visible-only $NO_2$ columns will be much larger than the old one.

In Fig. R2, we examine whether these effects show up in the BEHR data. Figure R2a shows the relative change in the visible-only $NO_2$ columns versus the ratio of $V_{\text{clear}}/V_{\text{cloudy}}$. The ratio $V_{\text{clear}}/V_{\text{cloudy}}$ sets a clear upper bound on the difference between the old and new visible-only $NO_2$ VCDs. What controls the ratio $V_{\text{clear}}/V_{\text{cloudy}}$ is shown in Fig. R2b. It increases rapidly as cloud pressure decreases, i.e. as the cloud hides more of the surface $NO_2$. When a large fraction of the $NO_2$ is near the surface, the effect is larger. This is illustrated by the fact that the top of the scatter in Fig. R2b has the greater surface $NO_2$ shape factor (here, the $NO_2$ mixing ratio divided by the column density). For a given cloud pressure, increasing the cloud fraction also increases $V_{\text{clear}}/V_{\text{cloudy}}$. All of these relationships are a natural result of clouds covering more $NO_2$.

In Fig. R2a, we also see that for a given ratio $V_{\text{clear}}/V_{\text{cloudy}}$, the magnitude of the difference between old and new BEHR $NO_2$ VCDs can vary quite significantly, depending primarily on the cloud radiance fraction. As the cloud radiance fraction decreases, the second term in both Eq. (R10) and Eq. (R11) becomes less important, so both become more similar to a clear-sky AMF and each other. However, at cloud radiance fractions near 1, the difference between old and new BEHR VCDs drops to 0. This happens because, as shown in Fig. R2c, when the cloud fractions are near 0 or 1, the geometric and radiance fractions converge, and for $f_g = f = 0$ or $f_g = f = 1$, Eq. (R10) and (R11) reduce to the same quantities.

To summarize, the conceptual difference is that the old AMF was a weighted sum of the clear and cloudy AMFs, but this did not account for the difference in magnitude between the to-ground and above cloud columns. The new AMF is a ratio of the expected slant column to the expected visible vertical column, which tends to include more $NO_2$ from the clear part of the pixel. Since OMI is

less sensitive overall to $NO_2$ in the clear part under most circumstances, the new AMFs are smaller, resulting in larger retrieved visible-only VCDs."

page14line21 What is UT?
Upper troposphere, we have defined it.

page15figure4 the (a) and (b) panels are not described anywhere.
We have added references in the appropriate sections (Sect. 4.3.2 and 4.4 respectively)

page15line3 I do not understand this hypothesis, since the profile shape itself depends not on the cloud information. The selection criteria of cloud fraction 0.2 only has impact on the NO2 column calculation, and it has no impact on the profile shape.
Our thinking was that, while the cloud fraction does not *directly* impact the profile shape, since clouds are correlated with lightning in the real world (and assuming that real world clouds correlated with modeled lightning), then selecting only cloud-free pixels would select for pixels whose daily profiles had less lightning contribution. In contrast, since the monthly profiles lack day-to-day variation, this would not affect them.
However, the first reviewer also found this section overly complex, so we have streamlined it, discussing only the actual conclusion: that the statistical distribution of UT $NO_x$ concentrations propagates differently through the monthly and daily profiles.

"

Ultimately, the fact that lightning is an intermittent but significant $NO_x$ source in the upper troposphere (UT) is the cause of this difference. Figure R3a shows the statistical distribution of $NO_2$ in the UT for two regions in the US: the southeast, which has significant lightning activity, and the northwest which has very little lightning. The distribution is highly skewed with a long tail in the southeast US due to the lightning activity, but not in the northwest US. Because of the nonlinear nature of the AMF calculation, this skewed distribution translates into different average VCD values.

Figure R3, panels b and c show average shape factors derived from monthly averaged and daily a priori profile for the southeast and northwest US. A shape factor is a profile divided by its integral:

$$S(p) = \frac{g(p)}{\int_{p_{\mathrm{surf}}}^{p_{\mathrm{trop}}} g(p)\,dp} \tag{R13}$$

A shape factor can be intepreted as the relative vertical distribution of $NO_2$. It appears implicitly in the AMF calculation (Eq. 2).

Here we see how the skewed UT $NO_2$ distribution affects the southeast US AMFs through the shape factor. Figure R3b shows that the statistically skewed UT $NO_2$ distribution causes shape factors calculated from the monthly average a priori profiles in the southeast US to have a larger fraction of the column $NO_2$ in the UT than that calculated from the daily profiles. Through Eq. (2), this leads to

[Figure]

Figure R3: **(a)** Frequency distribution (normalized to maximum) of average $NO_2$ above 400 hPa in the a priori profiles for the southeast and northwest US, from Jun–Aug 2012. **(b–c)** Mean a priori $NO_2$ shape factors over the southeast US (b) and northwest US (c) for Jun–Aug, 2012. Shape factors are defined as the $NO_2$ profile in mixing ratio divided by its integral in molec $cm^{-2}$. The error bars are $\pm 1\sigma$. The regions (SE and NW US) are shown in Fig. S4.

systematically greater AMFs (and therefore smaller VCDs) in the southeast when using the monthly profiles if the scattering weights ($w(p)$ in Eq. 2) are greater in the UT than near the surface, which is usually the case. In contrast, Fig. R3c shows no difference in the monthly or daily shape factors for the northwest US. For interested readers, a more mathematical argument is given in Sect. S2 of the supplement.

The implication is that, for regions with long-tailed statistical distributions of $NO_2$ concentrations, there will be systematic differences between a product using monthly average and daily a priori profiles. It is likely that the VCDs calculated using the daily a priori profiles are more accurate, because in theory daily a priori profiles should properly account for that long tail on days when it is relevant, whereas monthly profiles will average in the extreme values.

Finally we note that this difference between daily and monthly profiles may change in the future. Laughner et al. (2018) found that the simulation providing the $NO_2$ profiles had too much lightning in the southeast US. Correcting that may reduce the skewness of the UT $NO_2$ distribution. Work is underway to improve the representation of lightning for the southeast US $NO_2$ profiles. ”

page17line5 Is there probably a name or reference to this temperature profile?
Added citation to Bucsela et al. (2006)

page18line6-13 There are introductions of the previous method and why PSM method cannot be used here. It might be better to add a small introduction of the new CVM which is actually used in this study.
The concept of the old and new CVM methods are the same, only the implementation differs. We have clarified this:

> “BEHR v3.0A also uses a CVM gridding algorithm, however the implementation was changed. The new CVM algorithm is a slightly modified version of that provided by Kuhlmann et al. (2014), with a custom interface to allow communication between the Python code from `https://github.com/gkuhl/omi` and the BEHR Matlab code.”

page19line7 The ocean reflectance is calculated without MODIS data, therefore I do not understand the goal of this change to 460 nm. Even the impact is small, the re- flectance at 430 nm shall be used because of the reason exactly described in the text.
This could be argued either way; using the 430 nm reflectance would be more representative of the $NO_2$ fitting window, but using the 460 nm reflectance is consistent with the land reflectance wavelength, meaning that whatever small bias is imposed by using a wavelength on the edge of the $NO_2$ fitting window is consistent between the land and ocean. We have explained this more in the text:

> “In v3.0B, this was changed to be 460 nm, which is within the MODIS band used (459–479 nm). **While both approaches have merit, we chose to**

**move towards calculating the surface reflectance at similar wavelengths for consistency between the ocean and land data.** The change in VCD retrieved over ocean is very small..."

**References**

Acarreta, J. R., De Haan, J. F., and Stammes, P.: Cloud pressure retrieval using the O2-O2 absorption band at 477 nm, J. Geophys. Res. Atmos., 109, D05 204, doi:10.1029/2003JD003915, URL http://dx.doi.org/10.1029/2003JD003915, 2004.

Boersma, K. F., Vinken, G. C. M., and Eskes, H. J.: Representativeness errors in comparing chemistry transport and chemistry climate models with satellite UV–Vis tropospheric column retrievals, Geosci. Model Dev., 9, 875–898, doi:10.5194/gmd-9-875-2016, URL https://doi.org/10.5194/gmd-9-875-2016, 2016.

Bucsela, E., Celarier, E., Wenig, M., Gleason, J., Veefkind, J., Boersma, K., and Brinksma, E.: Algorithm for $NO_2$ vertical column retrieval from the ozone monitoring instrument, IEEE Trans. Geosci. Remote Sens., 44, 1245–1258, doi:10.1109/tgrs.2005.863715, URL https://doi.org/10.1109/tgrs.2005.863715, 2006.

Burrows, J. P. and Chance, K. V.: SCIAMACHY and GOME: the scientific objectives, in: Proc. SPIE, vol. 1715, doi:10.1117/12.140201, 1993.

Burrows, J. P., Richter, A., Dehn, A., Deters, B., Himmelmann, S., Voigt, S., and Orphal, J.: Atmospheric remote-sensing reference data from GOME–2. Temperature-dependent absorption cross section of $O_3$ in the 231-794 nm range, J. Quant. Spectrosc. Radiat. Transfer, 61, 509–517, 1999a.

Burrows, J. P., Weber, M., Buchwitz, M., Rozanov, V., Ladstätter-Weißenmayer, A., Richter, A., DeBeek, R., Hoogen, R., Bramstedt, K., Eichmann, K.-U., and Eisinger, M.: The Global Ozone Monitoring Experiment (GOME): Mission Concept and First Scientific Results, J. Atmos. Sci., 56, 151–175, doi:10.1175/1520-0469(1999)056⟨0151:TGOMEG⟩2.0.CO;2, 1999b.

Cahalan, R. F., Ridgway, W., Wiscombe, W. J., Gollmer, S., and Harshvardhan: Independent Pixel and Monte Carlo Estimates of Stratocumulus Albedo, J. Atmos. Sci., 51, 3776–3790, doi:10.1175/1520-0469(1994)051⟨3776:ipamce⟩2.0.co;2, URL https://doi.org/10.1175/1520-0469(1994)051<3776:ipamce>2.0.co;2, 1994.

Choi, S., Joiner, J., Choi, Y., Duncan, B. N., Vasilkov, A., Krotkov, N., and Bucsela, E.: First estimates of global free-tropospheric $NO_2$ abundances derived using a cloud-slicing technique applied to satellite observations from the Aura Ozone Monitoring Instrument (OMI), Atmos. Chem. Phys., 14, 10 565–10 588, doi:10.5194/acp-14-10565-2014, URL http://www.atmos-chem-phys.net/14/10565/2014/, 2014.

Emmons, L. K., Walters, S., Hess, P. G., Lamarque, J.-F., Pfister, G. G., Fillmore, D., Granier, C., Guenther, A., Kinnison, D., Laepple, T., Orlando, J., Tie, X., Tyndall, G.,

Wiedinmyer, C., Baughcum, S. L., and Kloster, S.: Description and evaluation of the Model for Ozone and Related chemical Tracers, version 4 (MOZART-4), Geosci. Model Dev., 3, 43–67, doi:10.5194/gmd-3-43-2010, URL `http://www.geosci-model-dev.net/3/43/2010/`, 2010.

Gorshelev, V., Serdyuchenko, A., Weber, M., Chehade, W., and Burrows, J. P.: High spectral resolution ozone absorption cross-sections—Part 1: Measurements, data analysis and comparison with previous measurements around 293 K, Atmos. Meas. Tech., 7, 609–624, doi:10.5194/amt-7-609-2014, URL `https://www.atmos-meas-tech.net/7/609/2014/`, 2014.

Jiang, Z., McDonald, B. C., Worden, H., Worden, J. R., Miyazaki, K., Qu, Z., Henze, D. K., Jones, D. B. A., Arellano, A. F., Fischer, E. V., Zhu, L., and Boersma, K. F.: Unexpected slowdown of US pollutant emission reduction in the past decade, PNAS, 115, 5099–5104, doi:10.1073/pnas.1801191115, URL `https://doi.org/10.1073/pnas.1801191115`, 2018.

Jin, Z., Charlock, T. P., Smith, W. L., and Rutledge, K.: A parameterization of ocean surface albedo, Geophys, Res, Lett,, 31, L22 301, doi:10.1029/2004GL021180, URL `http://dx.doi.org/10.1029/2004GL021180`, 2004.

Jin, Z., Charlock, T., Rutledge, K., Stamnes, K., and Wang, Y.: Analytical solution of radiative transfer in the coupled atmosphere-ocean system with a rough surface, Appl. Opt., 45, 7443–7455, 2006.

Kuhlmann, G., Hartl, A., Cheung, H. M., Lam, Y. F., and Wenig, M. O.: A novel gridding algorithm to create regional trace gas maps from satellite observations, Atmos. Meas. Tech., 7, 451–467, doi:10.5194/amt-7-451-2014, URL `https://www.atmos-meas-tech.net/7/451/2014/`, 2014.

Lamsal, L. N., Martin, R. V., van Donkelaar, A., Steinbacher, M., Celarier, E. A., Bucsela, E., Dunlea, E. J., and Pinto, J. P.: Ground-level nitrogen dioxide concentrations inferred from the satellite-borne Ozone Monitoring Instrument, J. Geophys. Res. Atmos., 113, D16 308, doi:10.1029/2007JD009235, URL `https://agupubs.onlinelibrary.wiley.com/doi/abs/10.1029/2007JD009235`, 2008.

Laughner, J. L., Zare, A., and Cohen, R. C.: Effects of daily meteorology on the interpretation of space-based remote sensing of $NO_2$, Atmos. Chem. Phys., 16, 15 247–15 264, doi:10.5194/acp-16-15247-2016, URL `http://www.atmos-chem-phys.net/16/15247/2016/`, 2016.

Laughner, J. L., Zhu, Q., and Cohen, R.: Evaluation of version 3.0B of the BEHR OMI $NO_2$ product, Atmos. Meas. Tech. Discuss., 2018, 1–25, doi:10.5194/amt-2018-248, URL `https://www.atmos-meas-tech-discuss.net/amt-2018-248/`, 2018.

Marais, E. A., Jacob, D. J., Choi, S., Joiner, J., Belmonte-Rivas, M., Cohen, R. C., Beirle, S., Murray, L. T., Schiferl, L., Shah, V., and Jaeglé, L.: Nitrogen oxides in the global upper troposphere: interpreting cloud-sliced $NO_2$ observations from the OMI satellite instrument, Atmos. Chem. Phys. Discuss., 2018, 1–14, doi:10.5194/acp-2018-556, URL `https://www.atmos-chem-phys-discuss.net/acp-2018-556/`, 2018.

Marshak, A., Davis, A., Cahalan, R., and Wiscombe, W.: Nonlocal independent pixel approximation: direct and inverse problems, IEEE Trans. Geosci. Rem. Sens., 36, 192–205, doi:10.1109/36.655329, URL `https://doi.org/10.1109/36.655329`, 1998.

Noel, S., Bovensmann, H., Burrows, J. P., Frerick, J., Chance, K. V., Goede, A. P. H., and Muller, C.: SCIAMACHY instrument on ENVISAT-1, in: Sensors, Systems, and Next-Generation Satellites II, edited by Fujisada, H., SPIE, doi:10.1117/12.333621, URL `https://doi.org/10.1117/12.333621`, 1998.

Pickering, K. E., Bucsela, E., Allen, D., Ring, A., Holzworth, R., and Krotkov, N.: Estimates of lightning $NO_x$ production based on OMI NO2 observations over the Gulf of Mexico, J. Geophys. Res. Atmos., 121, 8668–8691, doi:10.1002/2015JD024179, URL `http://dx.doi.org/10.1002/2015JD024179`, 2015JD024179, 2016.

Platnick, S., King, M., Wind, G., Ackerman, S., Menzel, P., and Frey, R.: MODIS/Aqua Clouds 5-Min L2 Swath 1km and 5km. NASA MODIS Adaptive Processing System, Goddard Space Flight Center, USA, doi:10.5067/MODIS/MYD06\_L2.006, 2015.

Russell, A., Perring, A., Valin, L., Bucsela, E., Browne, E., Min, K., Wooldridge, P., and Cohen, R.: "A high spatial resolution retrieval of $NO_2$ column densities from OMI: method and evalutation", Atmos. Chem. Phys., 11, 8543–8554, doi:10.5194/acp-11-8543-2011, 2011.

Russell, A. R., Valin, L. C., and Cohen, R. C.: Trends in OMI $NO_2$ observations over the United States: effects of emission control technology and the economic recession, Atmos. Chem. Phys., 12, 12 197–12 209, doi:10.5194/acp-12-12197-2012, 2012.

Ziemke, J., Chandra, S., and Bhartia, P.: Cloud slicing: A new technique to derive upper tropospheric ozone from satellite measurements, J. Geophys. Res. Atmos., 106, 9853–9867, 2001.